# The barley MLA13-AVR$_{A13}$ heterodimer reveals principles for immunoreceptor recognition of RNase-like powdery mildew effectors

Aaron W Lawson [ID][1], Andrea Flores-Ibarra[2], Yu Cao[1,2,3], Chunpeng An [ID][1], Ulla Neumann[4], Monika Gunkel [ID][2], Isabel M L Saur [ID][5,6], Jijie Chai [ID][1,2,3 ✉], Elmar Behrmann [ID][2 ✉] & Paul Schulze-Lefert [ID][1,6 ✉]

## Abstract

**Co-evolution between cereals and pathogenic grass mildew fungi is exemplified by sequence diversification of an allelic series of barley resistance genes encoding Mildew Locus A (MLA) nucleotide-binding leucine-rich repeat (NLR) immunoreceptors with an N-terminal coiled-coil domain (CNLs). Each immunoreceptor recognises a matching, strain-specific powdery mildew effector encoded by an avirulence gene (AVR$_a$). We present here the cryo-EM structure of barley MLA13 in complex with its cognate effector AVR$_{A13}$-1. The effector adopts an RNase-like fold when bound to MLA13 in planta, similar to crystal structures of other RNase-like AVR$_A$ effectors unbound to receptors. AVR$_{A13}$-1 interacts via its basal loops with MLA13 C-terminal leucine-rich repeats (LRRs) and the central winged helix domain (WHD). Co-expression of structure-guided MLA13 and AVR$_{A13}$-1 substitution variants show that the receptor–effector interface plays an essential role in mediating immunity-associated plant cell death. Furthermore, by combining structural information from the MLA13–AVR$_{A13}$-1 heterocomplex with sequence alignments of other MLA receptors, we engineered a single amino acid substitution in MLA7 that enables expanded effector detection of AVR$_{A13}$-1 and the virulent variant AVR$_{A13}$-V2. In contrast to the pentameric conformation of previously reported effector-activated CNL resistosomes, MLA13 was purified and resolved as a stable heterodimer from an in planta expression system. Our study suggests a common structural principle for RNase-like effector binding to MLAs and highlights the utility of structure-guided engineering of plant immune receptors for broadening their pathogen effector recognition capabilities.**

**Keywords** NLR Receptors; Pathogen Effectors; Co-evolution; Plant Immunity; Powdery Mildew
**Subject Category** Plant Biology

## Introduction

Plant–pathogen co-evolution involves reciprocal, adaptive genetic changes in both organisms, often resulting in population-level variations in nucleotide-binding leucine-rich repeat (NLR) immune receptors of the host and virulence-promoting effectors of the pathogen (Chisholm et al, 2006). NLRs often detect strain-specific pathogen effectors, so-called avirulence effectors (AVRs), inside plant cells, either by direct binding or indirectly by monitoring an effector-mediated modification of virulence targets (Maekawa et al, 2012). There are two main classes of modular sensor NLRs in plants, defined by a distinct N-terminal coiled-coil domain (CC; CNLs) or a Toll-Interleukin-1 Receptor (TIR; TNLs) domain, each of which plays a critical role in immune signalling after receptor activation (Chai et al, 2023; Saur et al, 2020). A subset of effector-activated sensor CNLs and TNLs engage additional 'helper NLRs' for immune signalling, some of which contain a HeLo-/RPW8-like domain or a CC at the N-terminus (Bentham et al, 2016; Collier et al, 2011). Immune signals initiated by activated sensor CNLs, sensor TNLs and helper NLRs converge on a rapid increase in $Ca^{2+}$ levels inside plant cells, often followed by host cell death, which is referred to as a hypersensitive response (HR) (Chai et al, 2023; Jones et al, 2024). In the two sensor CNLs *Arabidopsis thaliana* ZAR1 and wheat Sr35, effector-induced activation results in pentamerisation of heteromeric receptor complexes, called resistosomes, which is mainly mediated by oligomerisation of their central nucleotide-binding domains (NBDs) (Förderer et al, 2022; Wang et al, 2019a; Zhao et al, 2022). Recombinant ZAR1 and Sr35 resistosomes expressed in *Xenopus* oocytes exhibit non-selective cation channel activity, and the ZAR1 resistosome has additionally been shown to insert into planar lipid layers and display calcium-permeable cation-selective channel activity (Bi et al, 2021; Förderer et al, 2022). Thus, currently known structures of effector-activated sensor CNLs indicate the assembly of multimeric CNL resistosomes that mediate $Ca^{2+}$ influx in plant cells, ultimately leading to HR (Chai et al, 2023).

In the sister cereal species barley and wheat, numerous disease-resistance genes have been identified that encode CNLs conferring strain-specific immunity against the pathogenic grass powdery mildew fungi *Blumeria hordei* (*Bh*) or *Blumeria tritici* (*Bt*). Co-evolution with these Ascomycete pathogens has resulted in allelic

[1]Department of Plant Microbe Interactions, Max Planck Institute for Plant Breeding Research, 50829 Cologne, Germany. [2]University of Cologne, Faculty of Mathematics and Natural Sciences, Institute of Biochemistry, 50674 Cologne, Germany. [3]School of Life Sciences, Westlake University, 310031 Hangzhou, China. [4]Central Microscopy, Max Planck Institute for Plant Breeding Research, 50829 Cologne, Germany. [5]Institute for Plant Sciences, University of Cologne, 50674 Cologne, Germany. [6]Cluster of Excellence on Plant Sciences (CEPLAS), Max Planck Institute for Plant Breeding Research and University of Cologne, 50829 Cologne, Germany. ✉E-mail: chaijijie@westlake.edu.cn; elmar.behrmann@uni-koeln.de; schlef@mpipz.mpg.de

resistance specificities at some of these loci in host populations, with each resistance allele conferring immunity only to powdery mildew isolates expressing a cognate isolate-specific AVR effector (Bourras et al, 2019; Bourras et al, 2015; Manser et al, 2021; Praz et al, 2017; Seeholzer et al, 2010). The *Bh* avirulence effectors AVR$_{A1}$, AVR$_{A6}$, AVR$_{A7}$, AVR$_{A9}$, AVR$_{A10}$, AVR$_{A13}$, and AVR$_{A22}$ have been characterised and are recognised by the matching MLA receptors, MLA1, MLA6, MLA7, MLA9, MLA10, MLA13 and MLA22, respectively (Bauer et al, 2021; Lu et al, 2016; Saur et al, 2019a). Although these AVR$_A$s are unrelated at the sequence level, with the exception of allelic AVR$_{A10}$ and AVR$_{A22}$, structural predictions and the crystal structure of a *Bh* effector with unknown avirulence activity (CSEP0064) suggested that they share a common RNase-like scaffold with a greatly expanded and sequence-diversified effector family in the genomes of grass powdery mildew fungi, termed RNase-like associated with haustoria (RALPH) effectors (Bauer et al, 2021; Pedersen et al, 2012; Pennington et al, 2019; Seong and Krasileva, 2023). The crystal structures of *Bh* AVR$_{A6}$, AVR$_{A7}$-1, AVR$_{A10}$ and AVR$_{A22}$ validated this hypothesis and revealed unexpected structural polymorphisms between them that are linked to differentiation of RALPH effector subfamilies in powdery mildew genomes (Cao et al, 2023). The crystal structure of the RALPH effector AvrPm2a from *Bt*, detected by wheat CNL Pm2a, was also determined and belongs to a RALPH subfamily with 34 members, which includes *Bh* AVR$_{A13}$, *Bh* CSEP0064 and *Bt* E-5843 (Cao et al, 2023; Manser et al, 2021). For both barley MLA and wheat Pm2a, co-expression of matching receptor–avirulence pairs is necessary and sufficient to induce cell death in heterologous *Nicotiana benthamiana* (Bauer et al, 2021; Lu et al, 2016; Manser et al, 2021; Saur et al, 2019a). Similar to several other sensor CNLs, including ZAR1 and Sr35, mutations in MLA's MHD motif of the central NBD result in constitutive receptor signalling and effector-independent cell death (e.g., autoactive MLA10$^{D502V}$ and MLA13$^{D502V}$) (Bai et al, 2012; Crean et al, 2023; Maekawa et al, 2011). While yeast two-hybrid experiments and split-luciferase complementation assays indicate direct receptor–effector interactions for several matching MLA–AVR$_A$ pairs, similar assays suggest that wheat Pm2a indirectly detects AvrPm2 through interaction with the wheat zinc finger protein *Ta*ZF (Bauer et al, 2021; Manser et al, 2024; Saur et al, 2019a). The LRR of Pm2a mediates association with *Ta*ZF and recruits the receptor and AvrPm2a from the cytosol to the nucleus. However, the structural basis for how the MLA and Pm2 CNLs either directly or indirectly recognise RALPH effectors is lacking.

In this study, we used transient heterologous co-expression of barley MLA13 with its matching effector AVR$_{A13}$-1 in *N. benthamiana* leaves and affinity purification of heteromeric receptor complexes to demonstrate that the effector binds directly to the receptor. In contrast to the pentameric wheat Sr35 resistosome bound to AvrSr35 of *Puccinia graminis* f sp *tritici* (*Pgt*), we find that the MLA13–AVR$_{A13}$-1 heterocomplex is purified as a stable heterodimer and resolved using cryo-EM at a global resolution of 3.8 Å. Structural insights into the receptor–effector interface then served as a basis for structure-guided mutagenesis experiments. We co-expressed wild-type or mutant MLA13 and AVR$_{A13}$-1 in barley leaf protoplasts and heterologous *N. benthamiana* leaves to test the relevance of effector–receptor interactions revealed by the cryo-EM structure and their roles in immunity-associated cell death in planta. Combining structural

data with an in-depth sequence alignment between MLA receptors led to the identification of a single amino acid substitution in the MLA7 LRR that allows expanded RALPH effector detection.

# Results

## MLA13 $^{K98E/K100E}$-AVR$_{A13}$-1 purifies as a low-molecular-weight heterocomplex

We co-expressed N-terminal, GST-tagged MLA13$^{K98E/K100E}$ with C-terminal, Twin-Strep-tagged AVR$_{A13}$-1 in leaves of *N. benthamiana* via *Agrobacterium*-mediated transformation to facilitate the formation of potential receptor–effector heterocomplexes in planta, followed by affinity purification for structural studies. We observed that the substitutions MLA13$^{K98E/K100E}$, located in the CC domain, impede effector-triggered receptor-mediated cell death, therefore allowing us to express and purify these proteins without triggering cell death-mediated protein losses (Appendix Fig. S1). Notably, cell death was rescued when MLA13$^{K98E/K100E}$ was combined with the autoactive substitution D502V (MLA13$^{K98E/K100E/D502V}$; Appendix Fig. S1), suggesting that MLA13$^{K98E/K100E}$ may not impede the functionality of MLA13. Analogous substitutions were introduced in the helper CNL *At*NRG1.1, and demonstrated to impair its cell death activity and reduce association with the plasma membrane whilst retaining oligomerisation capability (Wang et al, 2023).

Affinity purification via the C-terminal, Twin-Strep-tag® on AVR$_{A13}$-1 resulted in the enrichment of both AVR$_{A13}$-1 and GST-MLA13$^{K98E/K100E}$ as demonstrated by SDS-PAGE analysis (Appendix Fig. S2A). A subsequent affinity purification via the GST tag on MLA13$^{K98E/K100E}$ resulted in the enrichment of MLA13$^{K98E/K100E}$ with concurrent co-purification of AVR$_{A13}$-1 (Appendix Fig. S2B), indicating that MLA13$^{K98E/K100E}$ and AVR$_{A13}$-1 formed a heterocomplex. Further analysis of the sample by size exclusion chromatography (SEC) revealed that the heterocomplex elutes at a volume implying a molecule significantly smaller than a hypothetical multimeric MLA13 resistosome (Fig. 1A). In line with the SEC results, negative stain transmission electron microscopy (TEM) analysis revealed homogeneous particles with a diameter of approximately 10 nm, suggesting a 1:1 heterodimer of MLA13 $^{K98E/K100E}$–AVR$_{A13}$-1 rather than multimeric resistosome assemblies (Fig. 1B). Notably, star-shaped particles characteristic of pentameric resistosome assemblies such as Sr35 were completely absent (Fig. 1B) (Förderer et al, 2022; Liu et al, 2022; Zhao et al, 2022).

Previously, structures of the pentameric Sr35 resistosome were determined after co-expression of wheat Sr35 with the avirulence effector AvrSr35 of the rust fungus *Pgt* in insect cell cultures and purification of a high-order resistosome from SEC (Förderer et al, 2022; Zhao et al, 2022). The unexpected observation of the heterodimeric assembly of MLA13$^{K98E/K100E}$ –AVR$_{A13}$-1 without detectable multimeric receptor–effector complexes from *N. benthamiana* prompted us to test whether co-expression of Sr35$^{L11E/L15E}$ with AvrSr35 in *N. benthamiana*, followed by the same purification method used for the purification of the MLA13$^{K98E/K100E}$ –AVR$_{A13}$-1 heterocomplex, leads to the formation of the Sr35 resistosome in planta (Appendix Fig. S3). SEC analysis of our in planta-purified Sr35 $^{L11E/L15E}$–AvrSr35 heterocomplex revealed an abundant high-order complex eluting with a calculated molecular

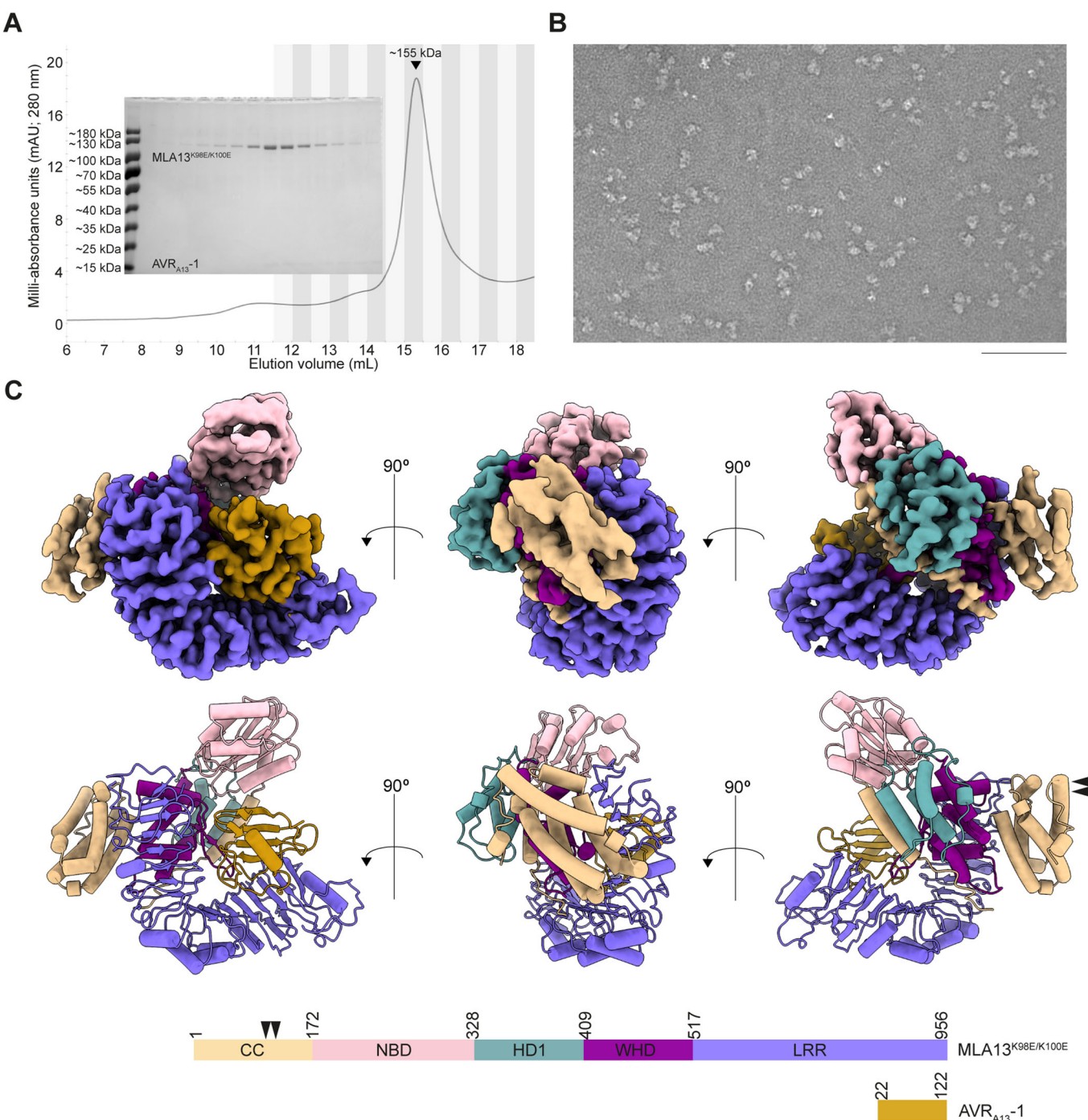

**Figure 1. The MLA13^{K98E/K100E}-AVR_{A13}-1 complex is purified and resolved as a heterodimer.**

(A) SEC profile of the N-terminal, GST-tagged MLA13^{K98E/K100E} in complex with C-terminal, Twin-Strep-HA-tagged AVR_{A13}-1 sample purified by a two-step affinity purification (Appendix Figure S2). Elution of the heterodimer (~155 kDa) is indicated by the peak at ~15.5 mL. Inset CBB-stained, SDS-PAGE gel presents fractions highlighted with alternating shades in the SEC profile. Samples run on a 12% gel. (B) Representative negative staining TEM image of the peak elution volume (~15.5 mL) from (A) diluted fivefold. Scale bar represents 100 nm. (C) Three orientations of the Coulomb potential map (above), atomic model (middle) and domain architecture (below) of the heterodimer. Arrows indicate the approximate locations of the MLA13^{K98E/K100E} substitutions introduced to impede in planta cell death. The workflow of cryo-EM data processing is presented as Fig. EV2. Source data are available online for this figure.

weight of ~875 kDa (Appendix Fig. S3B). Further negative stain TEM imaging of the corresponding SEC fraction confirmed a star-shaped complex that resembles the previously reported, insect cell-derived pentameric Sr35 resistosome (Appendix Fig. S3C)

(Förderer et al, 2022; Zhao et al, 2022). We then applied our protocol to the purification of a putative Sr50^{L11E/L15E} resistosome, an *Mla* ortholog in wheat (Appendix Fig. S4) (Chen et al, 2017). Co-expression of Sr50^{L11E/L15E} with *Pgt* AvrSr50 in *N. benthamiana* also

resulted in the elution of a high-order complex from SEC which was verified by negative stain TEM analysis to contain star-shaped particles similar to the Sr35 resistosome (Appendix Fig. S4C). This demonstrates that the formation of the putative Sr35 and Sr50 resistosomes is intrinsic to the co-expression of the two proteins, despite highly divergent expression systems in insect and plant cells. Moreover, successful purification of the putative Sr35 and Sr50 resistosomes suggests that our method may be applicable to other proteins such as MLA13–AVR$_{A13}$-1.

## Untagged MLA13 also forms a low-molecular-weight complex with AVR$_{A13}$-1

We then conducted additional purification experiments with various constructs to test if the non-pentameric, heterodimeric conformation MLA13$^{K98E/K100E}$ –AVR$_{A13}$-1 may be an artefact of the MLA13$^{K98E/K100E}$ substitutions or the N-terminal tag on MLA13 (Fig. EV1). We expressed and purified MLA13 and AVR$_{A13}$-1 without an N-terminal GST tag on MLA13 or with the same cell death-impeding substitutions used for the Sr35 and Sr50 resistosomes (L11E/L15E), all samples of which eluted at the same or later volume as the cryo-EM-resolved GST-MLA13$^{K98E/K100E}$–AVR$_{A13}$-1 heterodimer (Fig. EV1). Furthermore, we chose to use the cell death-impeding substitutions MLA13$^{K98E/K100E}$ for structural analysis because they resulted in higher protein yield compared to MLA13$^{L11E/L15E}$, the substitutions used for the purification of the Sr35 and Sr50 resistosomes, and because the cell death impediment caused by MLA13$^{K98E/K100E}$ is rescued when adding the autoactive substitution (MLA13$^{K98E/K100E/D502V}$; Appendix Fig. S1).

## Cryo-EM reveals the architecture of the MLA13–AVR$_{A13}$-1 heterodimer

Three independent MLA13$^{K98E/K100E}$ –AVR$_{A13}$-1 heterocomplex samples were prepared for cryo-EM analysis. During unsupervised 2D classification only a subset of identified particles yielded classes with features reminiscent of secondary structure elements. These had structures agreeing best with a heterodimeric but not with a pentameric assembly. Further classifying this subset of particles in 3D revealed heterodimeric complexes comprising one MLA13$^{K98E/K100E}$ and one AVR$_{A13}$-1. Reconstruction of these particles yielded a final cryo-EM Coulomb potential map with a global resolution of 3.8 Å (Figs. 1C and EV2). Local resolution analysis revealed that the core region of the complex, and importantly the interface between the receptor and AVR$_{A13}$-1, is defined up to 3.0 Å resolution. More peripheral regions such as the CC, the NBD and the first and last blades of the LRR show resolutions above 5.5 Å, implying their flexibility in the purified state of the heterodimer (Fig. EV2). Apart from these three regions, the quality of our map after machine learning-assisted sharpening was of sufficient quality to build an almost complete atomic model of the MLA13$^{K98E/K100E}$–AVR$_{A13}$-1 heterodimer.

The overall architecture of the MLA13$^{K98E/K100E}$–AVR$_{A13}$-1 heterodimer resembles a single effector-bound protomer of the pentameric Sr35 resistosome (Förderer et al, 2022; Zhao et al, 2022). While the resolution of the CC domain (MLA13$^{1-172}$) does not allow for fitting individual side chains, it clearly shows that the four amino-terminal alpha helices form a bundle reminiscent of the ligand-bound, monomeric Arabidopsis ZAR1–RKS1–PBL2$^{UMP}$

complex (Fig. 2A) (Wang et al, 2019b). Helix α3 is in close contact with a section of the MLA13 LRR (MLA1$^{518-956}$) that comprises a cluster of arginine residues (MLA13$^{R935/R936/R559/R561/R583/R612/R657/R703}$). This interdomain interaction is believed to be a precursor to the formation of the 'EDVID' motif-arginine cluster observed in the ZAR1 and Sr35 resistosomes following activation and CC rearrangement (Förderer et al, 2022; Zhao et al, 2022). The linker (MLA13$^{131-143}$) between helix α4A and the NBD (MLA13$^{173-328}$) lacks observable density, suggesting significant flexibility.

Similar to the CC domain, the quality of Coulomb potential map for the majority of the NBD does not allow for fitting individual side chains. In addition, the canonical nucleotide-binding site that is sequence-conserved with ZAR1 and Sr35 clearly lacks density for an ATP or ADP, similar to the ZAR1–RKS1–PBL2$^{UMP}$ complex (PDB: 6J5V) (Wang et al, 2019b). A motion-based deep generative model to investigate the flexibility remaining in the subpopulation of particles used for the 3D refinement implies that the NBD can sample a conformational space by rotating relative to the WHD (MLA13$^{410-517}$; Fig. 2B). Moreover, overlay of the MLA13 NBD after AVR$_{A13}$-1 binding to the receptor with the NBD of an AlphaFold3 model of the AVR$_{A13}$-1-bound MLA13 receptor shows conformational differences in NBD conformations between the prediction and experimental model (Fig. 2C; Appendix Fig. S5). Interestingly, a similar 'hinge' situated between the NBD and the WHD domain is observed when comparing the MLA13 NBD position to the NBD position in ZAR1 bound or unbound to the effector (Fig. 2B) (Wang et al, 2019b). Despite its flexibility, the MLA13 NBD does not, however, sample positions overlapping with the ZAR1 NBD, and the consensus position is about 75 degrees rotated compared to the ZAR1 resistosome assembly (Fig. 2B). Despite the differences observed for the NBD, the remaining domains of MLA13, namely HD1, WHD, and LRR, adopt positions similar to those observed in the non-resistosome ZAR1 structures (PDBs: 6J5W and 6J5V) (Wang et al, 2019b).

## AVR$_{A13}$-1 adopts an RNase-like fold and interacts with the LRR and WHD domains of MLA13

AVR$_{A13}$-1 adopts an RNase-like fold in planta reminiscent of the crystal structures reported for *E. coli*-expressed AVR$_{A6}$, AVR$_{A7}$-1, AVR$_{A10}$ and AVR$_{A22}$ of *Bh*, all of which share a structural core of two β-sheets and a central α-helix (Fig. 3A) (Cao et al, 2023). The N-terminal β-sheet consists of two antiparallel strands (β1 and β2), whilst the second β-sheet consists of four antiparallel β-strands (β3 to β6). Based on structural polymorphisms between *Bh* AVR$_{A6}$, AVR$_{A7}$-1, AVR$_{A10}$, AVR$_{A22}$ and *Bt* AvrPm2, AVR$_{A13}$-1 is most similar to *Bt* AVR$_{Pm2}$ and the structure of a *Bh* effector with unknown avirulence activity, CSEP0064 (Cao et al, 2023; Pennington et al, 2019). Each of the four crystallised AVR$_A$ effectors and *Bt* AvrPm2 share two conserved cysteine residues at the N and C termini, respectively, that form an intramolecular disulphide bridge connecting the N- and C-terminals. In AVR$_{A13}$-1, however, the position of the N-terminal cysteine is occupied by a leucine, impeding intramolecular disulphide formation with the C-terminal residue AVR$_{A13}$-1$^{C116}$. The conserved structural core of AVR$_{A13}$-1 and proximity of AVR$_{A13}$-1 N- and C-terminal ends show that intramolecular disulphide bridge formation is likely dispensable for the adoption of an RNase-like fold when bound to its receptor inside plant cells. This also indicates that binding to the receptor does not lead to extensive rearrangements of

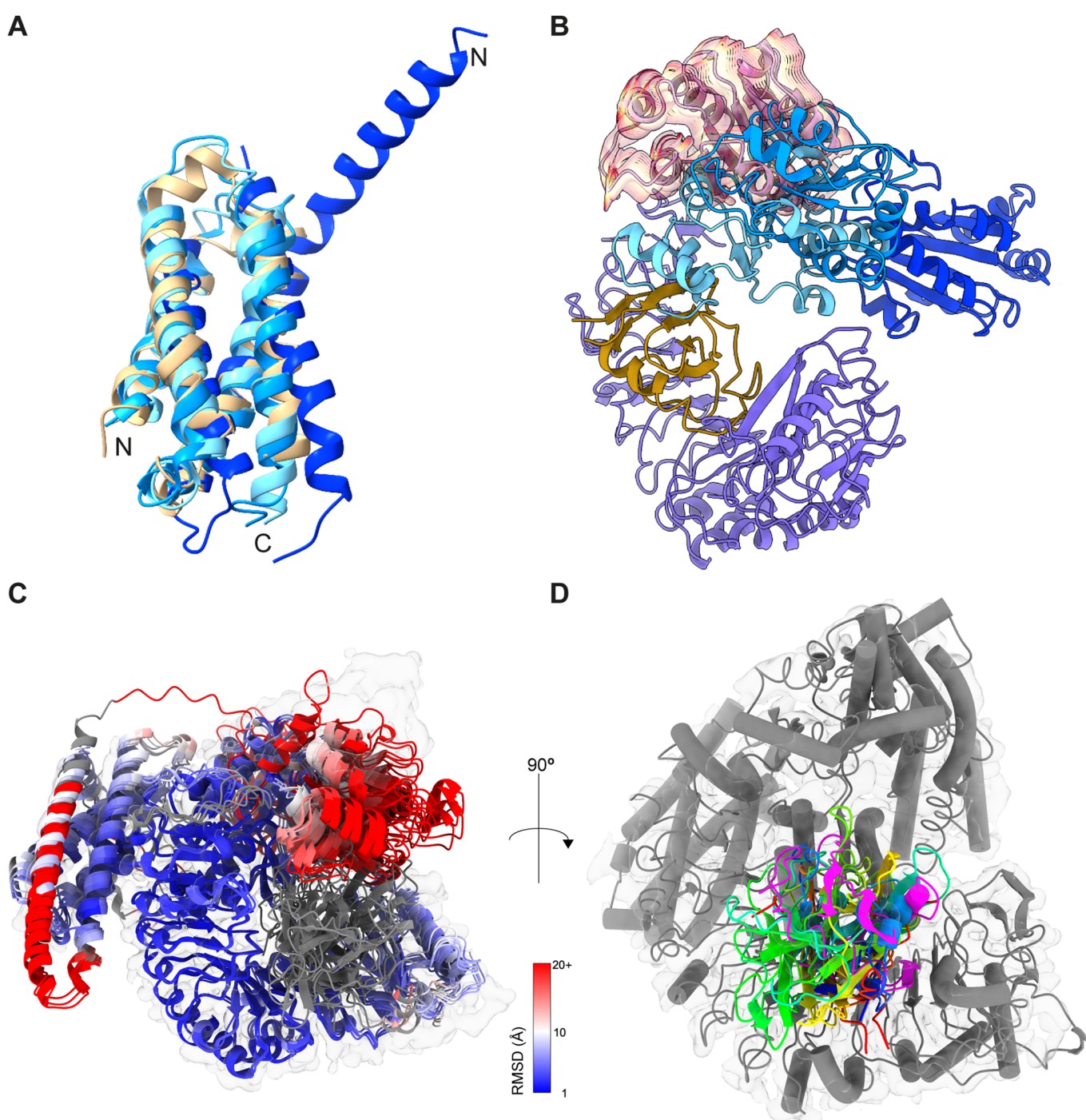

**Figure 2. Comparison of the MLA13^K98E/K100E-AVR_A13-1 heterodimer with ZAR1 and AlphaFold predictions.**

(A) Structural alignment of the CC domains of ZAR1–RKS1 (light blue; PDB: 6J5W), ZAR1–RKS1-PBL2^UMP (blue; PDB: 6J5V) and ZAR1 resistosome (dark blue; PDB: 6J5T) to the CC domain of the MLA13^K98E/K100E-AVR_A13-1 heterodimer (beige). (B) Structural alignment of ZAR1–RKS1 (light blue; PDB: 6J5W), ZAR1–RKS1-PBL2^UMP (blue; PDB: 6J5V) and ZAR1 resistosome (dark blue; PDB: 6J5T) to the MLA13^K98E/K100E-AVR_A13-1 heterodimer. Only the MLA13 NBD and LRR, AVR_A13-1 and NBDs of ZAR1 are shown. The red–yellow–red traces illustrate the major mode of conformational heterogeneity observed for the MLA13 NBD (average position shown in pink). (C) Top five models for the MLA13^K98E/K100E-AVR_A13-1 complex as predicted by AlphaFold 3 using five independent runs with different seeds. All five models were aligned to the MLA13 experimental atomic model and predicted MLA13 models are coloured by their RMSD deviation to the experimental model. Predicted parts of MLA13 not present in the experimental model and AVR_A13-1 model are coloured grey. The experimentally observed Coulomb potential map is shown in transparent grey. For all models, the position of the NBD does not align with the experimental model. In three predicted models the fourth helix of the CC bundle is too far elongated. (D) AlphaFold 3 predicts multiple orientations of AVR_A13-1 (coloured rainbow) that are all incorrectly rotated compared to the experimentally observed position (pink). The MLA13–AVR_A13-1 experimental Coulomb density map and model are shown in transparent grey and grey, respectively. For an overview of AlphaFold3 quality scores see Appendix Fig. S5.

the RNase-like fold compared to $AVR_A$ crystal structures of proteins purified from *E. coli* and unbound to their matching receptor (Cao et al, 2023).

The higher local resolution of the cryo-EM Coulomb potential map of the interface between the MLA13 LRR and $AVR_{A13}$-1 reveals interactions of the effector with multiple receptor residues, specifically from the concave side of the LRR and the WHD (Figs. 3A, 4A and EV3). To investigate the physiological relevance of the interactions between MLA13 and $AVR_{A13}$-1, we generated substitution variants of putative interacting residues in both the receptor and effector; we then transiently expressed these in barley protoplasts and leaves of *N. benthamiana* and tested for loss of $AVR_{A13}$-1-triggered and MLA13-mediated cell death (Figs. 3B,C,D and 4B,C,D).

Visualisation of the MLA13–$AVR_{A13}$-1 interface clarifies that the two basal loops of $AVR_{A13}$-1 ($AVR_{A13}$-1$^{47-74}$) play an essential role in the interaction with MLA13 and receptor-mediated cell death (Fig. EV3). Notably, the aromatic ring from $AVR_{A13}$-1$^{Y52}$ presents strong π–π stacking with MLA13$^{F900}$ and interacts with MLA13$^{Y934}$, an observation supported by a loss in cell death activity due to the single $AVR_{A13}$-1$^{Y52A}$ and MLA13$^{F900A}$ substitutions (Figs. 3, 4 and EV3). Contributing to stabilisation of the $AVR_{A13}$-1 basal loops and their interaction with the receptor, $AVR_{A13}$-1$^{F65}$ seemingly engages in a T-shaped interaction with the aromatic ring of MLA13$^{Y934}$. Furthermore, a notable reduction of cell death was observed when stacking the two substitutions $AVR_{A13}$-1$^{Y52A/G60A}$, presumably generating a steric clash between the backbone of $AVR_{A13}$-1$^{G60}$ and MLA13$^{Y491}$ (Figs. 3B,C and 4B,C). Reciprocally, the substitutions MLA13$^{Y491A}$ and MLA13$^{Y496A}$ in the WHD resulted in reduced cell death, suggesting that the WHD plays a critical role in triggering conformational changes in MLA13 that are necessary for cell death activity (Fig. 4B,C). Additional charged π interactions between MLA13$^{H643}$ and $AVR_{A13}$-1$^{N82}$ are also thought to be an important component of the receptor–effector interface. This is supported by the near-complete loss of cell death activity of the double substitution mutant MLA13$^{H643A/E936A}$ (Fig. 4B,C). We then tested the cell death activity of individual MLA13$^{E936A}$ and MLA13$^{S902A}$ variants (Fig. 4B,C). While MLA13$^{S902A}$ retained wild-type-like activity, the single receptor substitutions MLA13$^{F900A}$ and MLA13$^{E936A}$ resulted in a complete loss of cell death (Fig. 4B,C). Finally, we inferred that MLA13$^{S902}$ acts to stabilise MLA13$^{R938}$, an essential interactor of $AVR_{A13}$-1$^{D50}$ and $AVR_{A13}$-1$^{A51}$ that leads to a complete loss of cell death when introducing the single substitution MLA13$^{R938A}$ (Fig. 4B,C).

Co-immunoprecipitation (co-IP) assays were conducted for $AVR_{A13}$-1 interface substitution mutants that result in a reduced or loss of cell death activity (Appendix Fig. S6). Immunoprecipitation via the Twin-Strep-tag® on $AVR_{A13}$-1 interface mutants resulted in a decreased signal of MLA13, suggesting reduced receptor–effector interactions (Appendix Fig. S6).

## Expansion of MLA7 effector recognition specificity

Understanding the roles of receptor residues in the MLA13$^{K98E/K100E}$–$AVR_{A13}$-1 interface allowed us to generate a gain-of-function (GoF) MLA receptor based on amino acid sequence alignment with known MLA resistance specificities to *Bh* (Appendix Fig. S7) (Seeholzer et al, 2010). In this alignment, we observed that MLA7 is most similar to MLA13 with over 93% sequence conservation

among the two LRR domains (Appendix Fig. S7) (Cao et al, 2023). Closer inspection of the MLA7 and MLA13 sequence alignment revealed that only one of the LRR residues contributing to the MLA13$^{K98E/K100E}$–$AVR_{A13}$-1 interface was polymorphic between the two receptors at positions MLA7$^{L902}$ and the corresponding MLA13$^{S902}$ (Appendix Fig. S7). We then introduced the substitution MLA7$^{L902S}$ to test if this MLA13-mimicking receptor could gain detection of $AVR_{A13}$-1 while retaining the ability to detect its previously described cognate $AVR_{A7}$ effectors (Saur et al, 2019a). The co-expression of MLA7 WT with $AVR_{A7}$-2 in barley protoplasts results in a cell death response, whilst only weakly recognising $AVR_{A7}$-1, $AVR_{A13}$-1 and $AVR_{A13}$-V2, a virulent variant of $AVR_{A13}$-1 (Fig. 5A) (Crean et al, 2023; Lu et al, 2016). We then performed the same experiment with the MLA7$^{L902S}$ variant when not only was cell death activity retained upon co-expression with $AVR_{A7}$-2, but a gain of cell death activity was detected upon co-expression with $AVR_{A7}$-1, $AVR_{A13}$-1 and $AVR_{A13}$-V2 (Fig. 5A). Notably, MLA7$^{L902S}$ does not detect $AVR_{A22}$, indicating that GoF receptor detection could be limited to a subset of RALPH effectors (Fig. 5). The same co-expression experiments were performed in leaves of *N. benthamiana* with qualitatively similar results (Fig. 5B–D). MLA7$^{L902A}$ was also co-expressed with the same set of effectors in barley protoplasts with quantitatively similar results to those of MLA7$^{L902S}$ (Appendix Fig. S8).

## Discussion

Resolving the structure of the MLA13$^{K98E/K100E}$–$AVR_{A13}$-1 complex revealed a heterodimeric assembly that differs to two similar, albeit pentameric plant CNL resistosomes, *A. thaliana* ZAR1 and wheat Sr35 (Förderer et al, 2022; Wang et al, 2019a; Zhao et al, 2022). Similar structures such as monomeric ZAR1 are available (PDBs: 6J5W and 6J5V) and are believed to be intermediate forms of the effector-activated pentameric ZAR1 resistosome (Wang et al, 2019a, 2019b). The ZAR1–RKS1 complex binds ADP, and subsequent PBL2$^{UMP}$ binding in the presence of ATP results in allosteric changes, allowing the exchange of ADP to ATP in the NBD and the formation of a fully activated ZAR1 resistosome (Wang et al, 2019a, 2019b). ZAR1–RKS1 binding of PBL2$^{UMP}$ in the absence of ATP results in a nucleotide-free, ligand-bound intermediate complex (PDB: 6J5V), a conformation reminiscent of the MLA13$^{K98E/K100E}$–$AVR_{A13}$-1 heterodimer.

The heterodimeric assembly of the MLA13$^{K98E/K100E}$–$AVR_{A13}$-1 complex is unexpected and prompts questions regarding the potential absence of a multimeric resistosome as reported for the CNLs ZAR1 and Sr35 (Förderer et al, 2022; Wang et al, 2019a; Zhao et al, 2022). We propose three possible explanations for the stable MLA13$^{K98E/K100E}$–$AVR_{A13}$-1 heterodimeric assembly. First, a hypothetical MLA13 resistosome may disassemble into receptor–effector heterodimers upon cell lysis and protein purification or is otherwise inaccessible to our purification method. Second, the concentration of a hypothetical MLA13 resistosome is too low compared to the heterodimers to be isolated by our purification protocol, but still sufficient for receptor-mediated cell death triggered by the effector. Third, the stable MLA13$^{K98E/K100E}$–$AVR_{A13}$-1 heterodimer represents a non-pentameric, novel immunostimulatory output. Of note, regardless of whether the MLA13$^{K98E/K100E}$–$AVR_{A13}$-1 heterodimer results from disassembly of a hypothetical MLA13 resistosome or is a ligand-bound intermediate state receptor with immunostimulatory activity, the

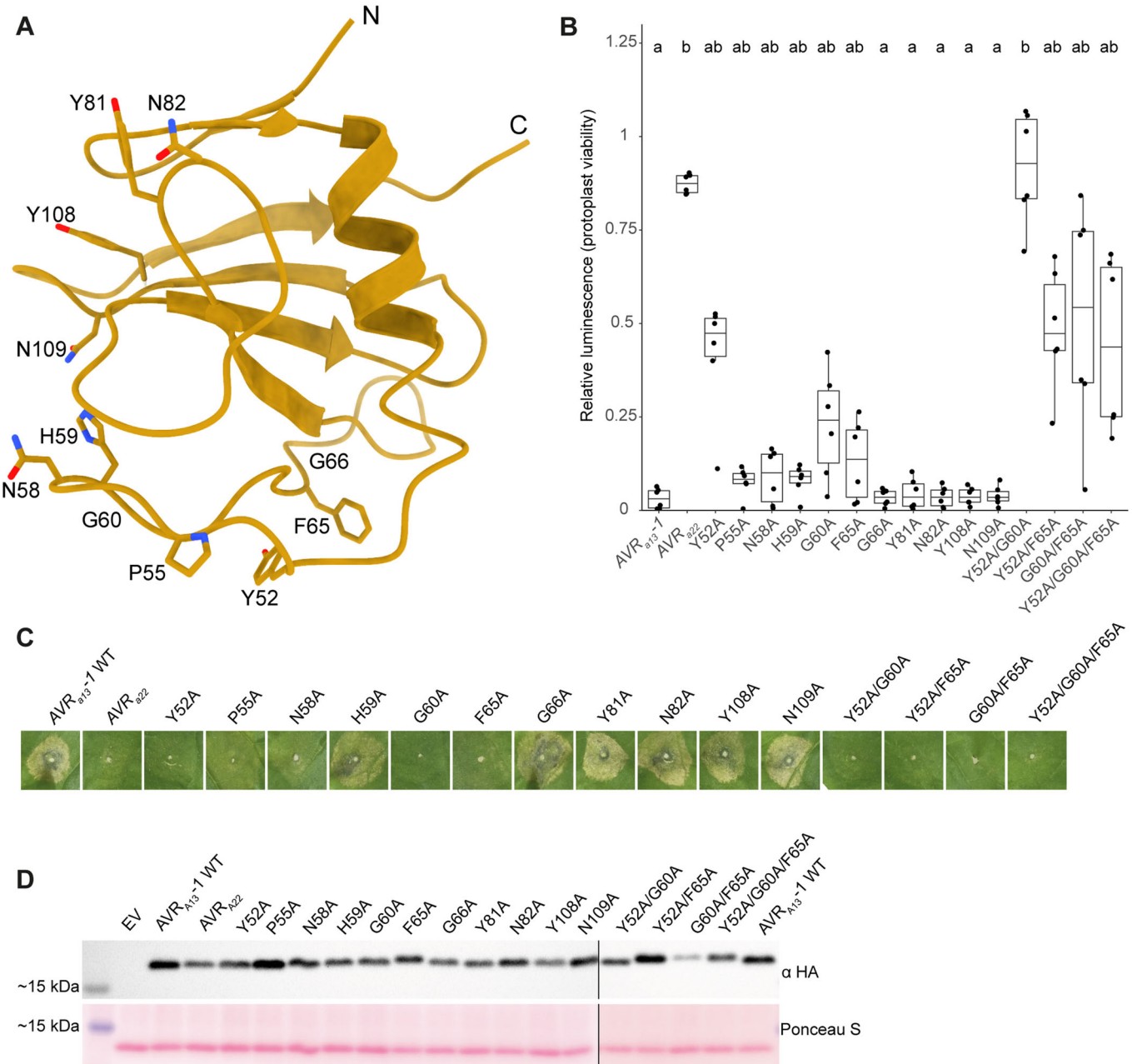

**Figure 3. The AVR$_{A13}$-1 basal loops are primarily responsible for interacting with the MLA13 LRR.**

(A) Cryo-EM-derived atomic model of AVR$_{A13}$-1 with side chains shown for residues predicted to contribute to the interface with MLA13 and experimentally tested for loss of MLA13-mediated cell death. (B) Co-expression of MLA13 with AVR$_{A13}$-1 substitution mutants in barley protoplasts. Luminescence is normalised to EV + MLA13 ( = 1). High relative luminescence suggests low cell death response. Six data points represent two technical replicates performed with three independently prepared protoplast samples. Treatments labelled with different letters differ significantly ($P < 0.05$) according to the Dunn's test ($P$ values are included as Source Data for Fig. 3). In the box plot, the top, middle, and bottom horizontal lines of the box correspond to the upper quartile, the median, and the lower quartile, respectively. The whiskers extend to the smallest and largest data points within 1.5 times the interquartile range from Q1 and Q3. Any points outside this range are plotted as dots and considered outliers. (C) *Agrobacterium*-mediated co-expression of MLA13 with AVR$_{A13}$-1 interface substitution mutants in leaves of *N. benthamiana*. Three independent replicates were performed with two *Agrobacterium* transformations and plant batches (Source Data for Fig. 3). (D) Western blot analysis of AVR$_{A13}$-1 substitution mutants. Samples were run on a 10% gel. Source data are available online for this figure.

observed receptor–effector interface is necessary for MLA13-mediated cell death.

This interface is primarily mediated by interactions supported by residues in the MLA13 WHD, LRR and two basal loops in AVR$_{A13}$-1. Similarly, earlier structure–function analyses of AVR$_{A10}$, AVR$_{A22}$ and

AVR$_{A6}$ hybrid effectors suggested that multiple highly polymorphic effector surface residues in the basal loops of each of these *Bh* RALPH effectors are indispensable for recognition by their matching MLA receptors (Bauer et al, 2021; Cao et al, 2023). This indicates the existence of a common structural principle by which functionally diversified MLA

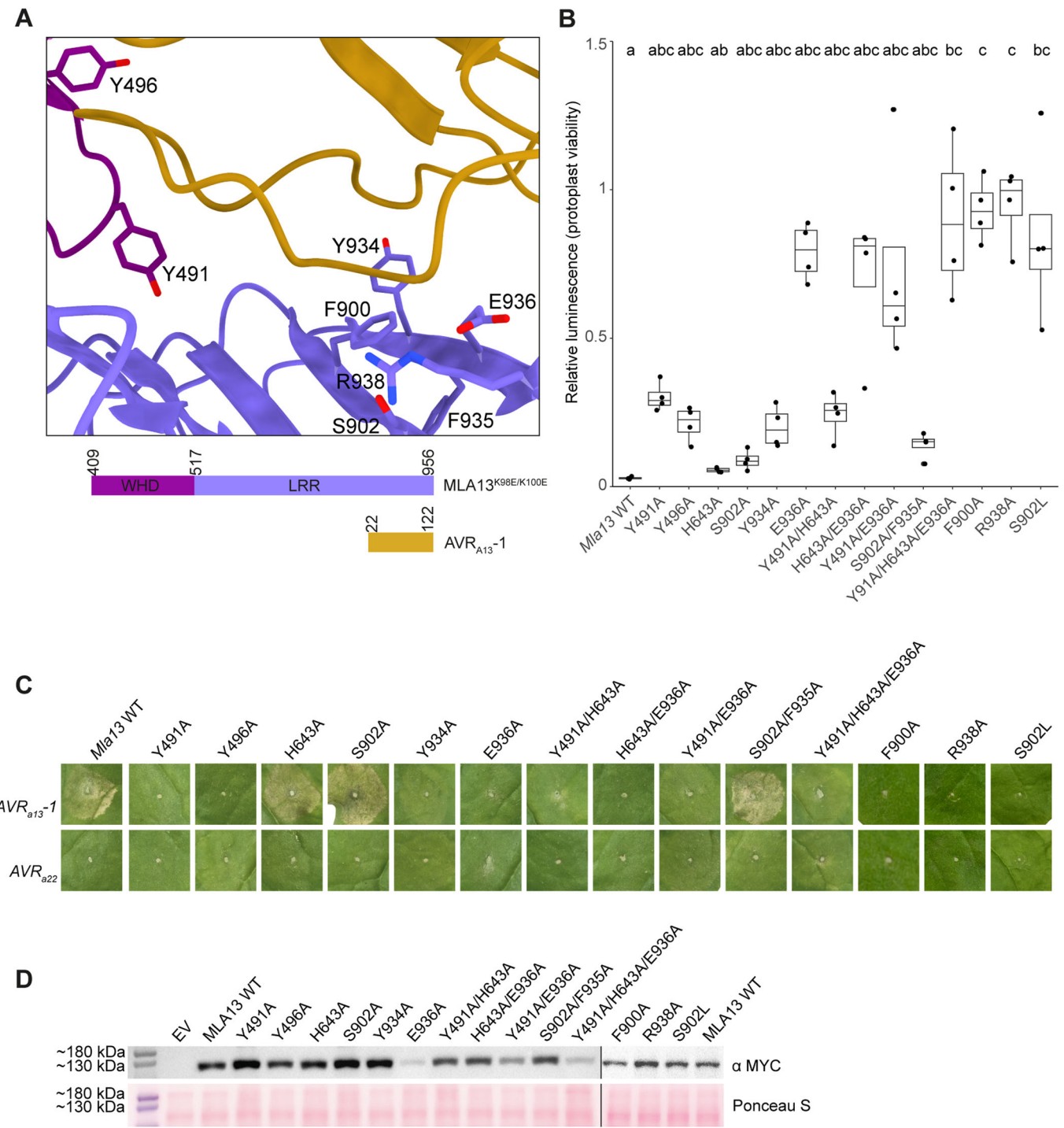

**Figure 4. Minimal but spatially distributed recognition of AVR_A13-1 by the MLA13 LRR and WHD.**

(A) The MLA13–AVR_A13-1 interface exhibiting MLA13 residues that were experimentally shown to contribute to AVR_A13-1-mediated cell death. (B) Co-expression of AVR_A13-1 with MLA13 substitution mutants in barley protoplasts. Each MLA13 variant was normalised to its own autoactivity; luminescence is normalised to EV + MLA13 variant (= 1). High relative luminescence suggests low cell death response. The four data points represent two technical replicates performed with two independently prepared protoplast samples. Treatments labelled with different letters differ significantly (*P* < 0.05) according to the Dunn's test (*P* values are included as Source Data for Fig. 4). In the box plot, the top, middle, and bottom horizontal lines of the box correspond to the upper quartile, the median, and the lower quartile, respectively. The whiskers extend to the smallest and largest data points within 1.5 times the interquartile range from Q1 and Q3. Any points outside this range are plotted as dots and considered outliers. (C) *Agrobacterium*-mediated co-expression of AVR_A13-1 with MLA13 substitution mutants believed to contribute to MLA13 interface and cell death response in leaves of *N. benthamiana*. Three independent replicates were performed with two *Agrobacterium* transformations and plant batches (Source Data for Fig. 4). (D) Western blot analysis of MLA13 substitution mutants. Samples were run on a 12% gel. Source data are available online for this figure.

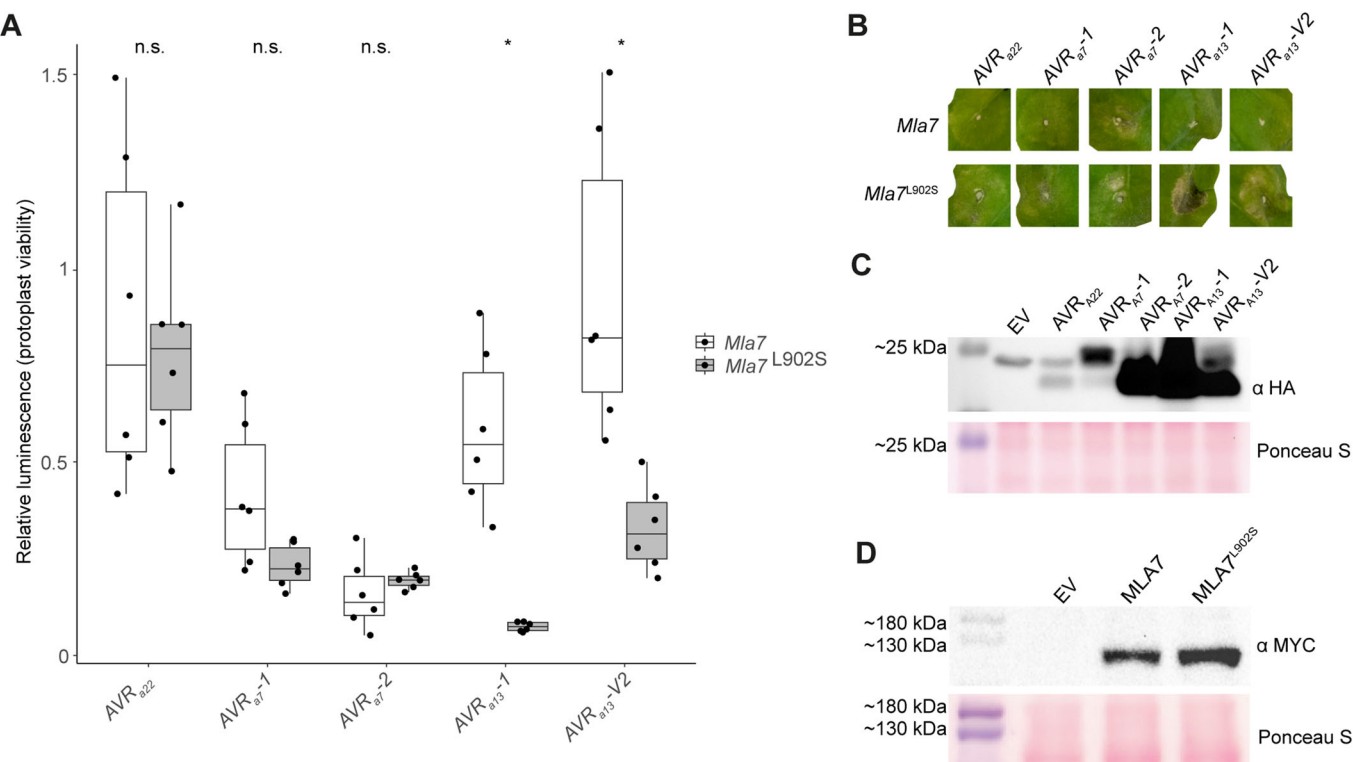

**Figure 5. The MLA7^L902S substitution mutant results in expanded effector recognition.**

(A) Co-expression of MLA7 and MLA7^L902S with AVR_A7 and AVR_A13 variants in barley protoplasts. Luminescence is normalised to EV + MLA7 ( = 1) or EV + MLA7^L902S ( = 1). High relative luminescence suggests low cell death response. The six data points represent two technical replicates performed with three independently prepared protoplast samples. Treatments labelled with an asterisk differ significantly ($P < 0.05$) according to the Welch two-sample $t$ test. The $P$ values for AVR_a22, AVR_a7-1, AVR_a7-2, AVR_a13-1 and AVR_a13-V2 are 0.682, 0.059, 0.386, 0.002 and 0.010, respectively. In the box plot, the top, middle, and bottom horizontal lines of the box correspond to the upper quartile, the median, and the lower quartile, respectively. The whiskers extend to the smallest and largest data points within 1.5 times the interquartile range from Q1 and Q3. Any points outside this range are plotted as dots and considered outliers. (B) *Agrobacterium*-mediated co-expression of MLA7 and MLA7^L902S with AVR_A7 and AVR_A13 variants in *N. benthamiana* leaves. Three independent replicates were performed (Source Data for Fig. 5). (C) Western blot analysis of the effector variants tested in (B). (D) Western blot analysis of MLA7 and MLA7^L902S. Samples were run on a 12% SDS-PAGE gel. Source data are available online for this figure.

receptors recognise sequence-unrelated RALPH effectors via their polymorphic basal loops (Fig. EV4). This is consistent with the observation that the structural core of RALPH effectors with two β-sheets and a central α-helix of AVR_A13-1 does not directly contribute to binding MLA13. Interestingly, AlphaFold3 generated several models in which AVR_A13-1 binds to the LRR domain of MLA13, but neither the binding site to the LRR nor the orientation of the effector relative to the LRR corresponds to the experimentally determined receptor-effector interface (Fig. 2D). Why would MLA receptors preferentially recognise AVR_A effectors at the basal loops and not at other distant surface regions of the RNase-like scaffold? We hypothesise that the polymorphic sequences in the basal loops are important for the virulence activity of these *Bh* RALPH effectors that lack catalytic activity and may allow them to interact with different virulence targets. This differs from the first structure of an LRR protein, the ribonuclease inhibitor protein, in complex with ribonuclease A; here, direct protein-protein contacts within the active site revealed a competitive mode of inhibition (Kobe and Deisenhofer, 1996). In addition, wheat CNL Pm2a is believed to detect the *Bt* RALPH effector AvrPm2a on the opposite effector side, termed the 'head epitope' which comprises the juxtaposed N- and C termini (Manser et al, 2021). This could be explained by the

finding that Pm2a recognises AvrPm2a indirectly through interaction with the wheat zinc finger protein TaZF[23]. An alternative hypothesis is that MLAs avoid recognising conserved structural elements, such as those of RNase-like scaffolds, to prevent interacting with RNase-like host proteins that may trigger a cell death in the absence of pathogens.

Here we provide evidence that residues in the C-terminal region of the MLA13 LRR are essential for receptor-mediated cell death activation upon detection of its cognate effector AVR_A13-1. The broader relevance of the C-terminal LRR region among MLA receptors for the detection of different AVR_A effectors is supported by domain swap experiments between LRR regions of MLA1 and MLA6 and MLA10 and MLA22, respectively (Bauer et al, 2021; Shen et al, 2003). Our results show that although the LRR region is the most polymorphic among characterised MLA receptors, there are relatively few polymorphic residues in the MLA13 LRR that are critical for recognition of AVR_A13-1 (Seeholzer et al, 2010). This information, combined with knowledge of natural LRR sequence polymorphisms among MLA receptors with distinct AVR_A effector recognition specificities, has informed the design of an MLA receptor with only a single-base edit (MLA7^L902S). Importantly, in the context of MLA13, substitution of MLA13^S902A resulted in a

retention of AVR$_{A13}$-1-triggered cell death activity, suggesting that MLA13$^{S902}$ may not play a critical role in supporting the interface with AVR$_{a13}$-1. In the context of MLA7, the MLA7$^{L902S}$ substitution is crucial for a gain of AVR$_{a13}$-1 detection, suggesting that the bulky MLA7$^{L902}$ disrupts the stability of MLA7$^{R938}$ and its essential role in effector interaction. Nevertheless, without experimental structures of MLA7$^{L902S}$ bound to AVR$_{A13}$-1 and AVR$_{A7}$-2, we cannot rule out the possibility that variation in the basal loop lengths of these two AVR$_A$ effectors might lead to conformationally different receptor–effector interfaces (Appendix Fig. S9). In fact, the structural polymorphisms between the two RALPH subfamilies, which include AVR$_{A7}$-2 and AVR$_{A13}$-1, differ primarily in the lengths of the four antiparallel β-strands (β3 to β6) of the second β-sheet and not the number of structural elements, thereby resulting in different lengths of the basal loops (Cao et al, 2023).

Expanding effector detection specificity by minimal perturbations such as single-base gene editing is an attractive approach for accomplishing more durable disease resistance in crops. Characterised *Mla* resistance specificities to *Bh* are alleles of one of three highly sequence-diverged CNL homologues at the complex *Mla* locus (Halterman and Wise, 2004; Maekawa et al, 2019; Wei et al, 2002). This precludes the generation of lines expressing two or more homozygous *Mla* resistance specificities by crossings between accessions encoding naturally polymorphic *Mla*s. The expanded detection capability of MLA7$^{L902S}$ is a promising and notable proof-of-principle, as the receptor is able to recognise multiple RALPH effectors belonging to two phylogenetic subfamilies. The new repertoire of matching effectors detected by MLA7$^{L902S}$ is simultaneously expressed in several globally distributed *Bh* strains and includes the virulent effector, AVR$_{A13}$-V2, which is presumed to be the result of resistance escape of MLA13 due to selection pressures (Crean et al, 2023; Lu et al, 2016; Saur et al, 2019a). Furthermore, certain allelic *Pm3* resistance specificities in wheat confer both strain-specific immunity to *Bt* and non-host resistance to other cereal mildews (Bourras et al, 2019). These wheat Pm3 CNL receptors recognise strain-specific matching *Bt* RALPH effectors and conserved RALPH effector homologues in rye mildew (*B. graminis* f sp *secale*), thereby restricting growth of rye mildew on wheat (Bourras et al, 2019). Given that barley MLA7$^{L902S}$ also confers enhanced cell death activity to the naturally occurring virulent variant of AVR$_{A13}$-1, AVR$_{A13}$-V2, and that the 34 members of this RALPH subfamily include several *Bt* effectors, including AvrPm2a and *Bt* E-5843, it seems possible that this or other engineered MLA receptors could enhance barley non-host resistance to other cereal mildews (Cao et al, 2023; Lu et al, 2016; Manser et al, 2021). Future work will complement our findings by generating gene edited barley lines expressing synthetic MLAs for resistance testing.

# Methods

### Reagents and tools table

| Reagent/resource | Reference or source | Identifier or catalogue number |
|---|---|---|
| **Experimental models** | | |
| DH5α cells (*E. coli*) | Invitrogen™ | Cat. no. 18265017 |
| Electrocompetent GV3101 (pMP90) cells (*A. tumefaciens*) | GoldBio | Cat. no. CC-207-5×50 |
| Wild-type plants (*N. benthamiana*) | Not applicable | Not applicable |

| Reagent/resource | Reference or source | Identifier or catalogue number |
|---|---|---|
| **Recombinant DNA** | | |
| pGWB402SN | Addgene | Cat. no. 224551 |
| pGWB402SC | Addgene | Cat. no. 224552 |
| pGWB424 | Addgene | Cat. no. 74818 |
| pGWB517 | Addgene | Cat. no. 74859 |
| *Mla13* (codon altered for expression in *S. frugiperda*) | GeneArt (Thermo Fisher Scientific Inc.) | Not applicable |
| *AVR$_{a13}$-1* (codon altered for expression in *N. benthamiana*) | GeneArt (Thermo Fisher Scientific Inc.) | Not applicable |
| *AVR$_{a13}$-V2* (codon altered for expression in *N. benthamiana*) | GeneArt (Thermo Fisher Scientific Inc.) | Not applicable |
| *AVR$_{a22}$* (codon altered for expression in *N. benthamiana*) | GeneArt (Thermo Fisher Scientific Inc.) | Not applicable |
| *AVR$_{a7}$-1* (codon altered for expression in *N. benthamiana*) | GeneArt (Thermo Fisher Scientific Inc.) | Not applicable |
| *AVR$_{a7}$-2* (codon altered for expression in *N. benthamiana*) | GeneArt (Thermo Fisher Scientific Inc.) | Not applicable |
| **Antibodies** | | |
| HA-Tag (C29F4) Rabbit mAb | Cell Signaling Technology® | Cat. no. 3724 |
| Myc Tag mAb | Invitrogen™ | Cat. no. R950-25 |
| Anti-rabbit IgG, HRP-linked Antibody | Cell Signaling Technology® | Cat. no. 7074 |
| Rabbit Anti-Mouse IgG H&L (HRP) | Abcam | Cat. no. ab6728 |
| **Chemicals, enzymes, and other reagents** | | |
| Gateway™ LR Clonase™ II Enzyme Mix | Invitrogen™ | Cat. no. 11791020 |
| Luciferase Assay System | Promega | Cat. no. E1500 |
| Protease Inhibitor Mix P | SERVA | Cat. no. 39103.03 |
| BioLock | IBA Lifesciences | Cat. no. 2-0205-050 |
| Strep-Tactin® XT Sepharose chromatography resin | Cytiva | Cat. no. 29401324 |
| Glutathione Sepharose 4B GST-tagged protein purification resin | Cytiva | Cat. no. 17075601 |
| **Software** | | |
| EPU | Thermo Fisher Scientific Inc. | Version 2.12 |
| CryoSPARC | Structura Biotechnology Inc. | Version 4.1.1 +patch 240110 |
| Phenix | Liebschner et al (2019) | Version 1.21-5207 |
| ChimeraX | UCSF RBVI | Version 1.7 |
| **Other** | | |
| Luminescence plate reader | Berthold Technologies | Centro XS³ LB 960 |

| Reagent/resource | Reference or source | Identifier or catalogue number |
|---|---|---|
| Superose™ 6 Increase small-scale SEC column | Cytiva | Cat. no. 29091596 |
| TEM Grids, Formvar/Carbon Film-coated, F 10 nm/C 1 nm, 400 Mesh, Cu, 50 | Science Services | Cat. no. EFCF400-Cu-50 |
| TEM Grids, Graphene Oxide on Quantifoils R2/4, 200 Mesh, Cu, 50 pieces | Science Services | Cat. no. ERGOQ200R24Cu50 |
| Vitrobot | Thermo Fisher Scientific Inc. | Mark IV |
| Transmission electron microscope | Hitachi | HT7800 with EMSIS XAROSA camera |
| Cryo-electron microscope | Thermo Fisher Scientific Inc. | Titan Krios G3i |

## Plant growth

Seeds of wild-type *N. benthamiana* were sown in peat-based potting soil with granulated cork on the surface to prevent pest infestation. Daily irrigation solution contained an electrical conductivity of 2.2 and a mixture of macro and micro nutrients. A photoperiod of 16 h was used with broad-spectrum LED lights emitting 220 μmol/m²/s supplemented by ambient sunlight.

Barley protoplasts isolated from Golden Promise seedlings that were grown on peat-based potting soil at 19 °C and 70% humidity for 7–9 days.

## Transient transformation of *N. benthamiana* for recombinant protein expression and purification

The coding sequences of *Mla13* containing a stop codon was transferred from pDONR221 using Gateway™ LR Clonase™ into pGWB424 containing an N-terminal fusion GST tag in the vector backbone. $AVR_{a13}$-1 without a stop codon was transferred from pDONR221 using Gateway™ LR Clonase™ into pGWB402SC containing a C-terminal Twin-Strep-tag® followed by a single HA tag in the vector backbone. Both constructs were individually electroporated into *Agrobacterium tumefaciens* strain GV3101::pMP90RK and selected on plates of Luria/Miller (LB) broth with agar containing spectinomycin (100 μg/mL), gentamycin (25 μg/mL), rifampicin (50 μg/mL) and kanamycin (25 μg/mL) and grown for two days at 28 °C. Three colonies were picked and cultured overnight in a 10-mL liquid LB starter culture with the above antibiotics at 28 °C. Two millilitres of the starter culture were added to and cultured in 350 mL of liquid LB broth containing the above antibiotics for 14 h at 28 °C. The cultures were pelleted at 4000 RCF for 15 min and resuspended in infiltration buffer (10 mM MES (pH 5.6), 10 mM $MgCl_2$, 500 μM acetosyringone) to an $OD_{600}$ of 2 for each construct. The bacterial suspensions were combined at a 1:1 ratio and infiltrated into leaves of 4-week-old *N. benthamiana* plants. The infiltrated plants were stored in the dark for 24 h before they were returned to normal growth conditions where they grew for an additional 24 h. The leaves were frozen in liquid nitrogen and stored at −80 °C until they were processed.

## Protein purification for cryo-EM

One hundred grams of transiently transformed *N. benthamiana* leaf tissue were ground in a prefrozen mortar and pestle and gradually added to 200 mL of lysis buffer (buffer A; 50 mM Tris-HCl (pH 7.4), 150 mM NaCl, 5% glycerol, 10 mM DTT, 0.5% polysorbate 20, two vials of protease inhibitor cocktail, 5% BioLock; pH adjusted to 7.4) until the lysate was defrosted and at 4 °C. The lysate was split into two 250 mL centrifuge bottles, centrifuged twice at 30,000 RCF for 15 min and filtered through double-layered miracloth after each centrifuge run.

Five hundred microlitres of Strep-Tactin® XT Sepharose resin (Cytiva) were equilibrated in wash buffer (buffer B; 50 mM Tris-HCl (pH 7.4), 150 mM NaCl, 2 mM DTT, 0.1% polysorbate 20; pH adjusted to 7.4). The resin was added to the lysate and incubated by end-over-end rotation at 4 °C for 30 min. The resin was washed three times with buffer B and finally isolated in a 1.5-mL tube. Five hundred microlitres of Strep-Tactin XT Sepharose resin elution buffer (buffer C; buffer B supplemented with 50 mM biotin; pH adjusted to 7.4) was added to the resin and rotated end-over-end for 30 min. The above elution step was repeated five times.

The five eluates were centrifuged at 16,000 RCF for 1 min and 450 μL of supernatant were removed from each eluate and pooled. Two hundred microlitres of Glutathione Sepharose 4B resin (Cytiva) was equilibrated in buffer B and added to the Strep-Tactin® XT eluate was combined with the Glutathione Sepharose 4B resin and incubated by mixing end-over-end for 2 h at 4 °C. The Glutathione Sepharose 4B resin was washed twice before with buffer B. Elution from the Glutathione Sepharose 4B resin was performed by adding 200 μL of buffer D (buffer B supplemented with 50 mM reduced glutathione; pH adjusted to 7.4) and rotated end-over-end for 30 min. Elution was repeated for a total of three times. The four eluates were centrifuged at 16,000 RCF for 1 min and 150 μL of supernatant were removed from each eluate. Twenty microlitres from the first eluate were used for cryo-EM grid preparation and the remaining eluate(s) were pooled and analysed by SEC.

For SEC, a Superose 6 increase 10/300 GL column (Cytiva) was equilibrated with buffer B. Five hundred microlitres of the pooled GST eluate were loaded into the column and run at 0.3 mL/min. Forty-five microlitres of the 500 μL fractions were loaded on SDS-PAGE gels.

The Sr35 and Sr50 resistosomes were purified from 100 g of leaf tissue with the above method. The in planta cell death activity was abrogated for purification purposes through the introduction of the L11E/L15E substitutions in the receptors. The Sr35 resistosome was purified by a single-step affinity co-immunoprecipitation via the C-terminal Twin-Strep-tag® on AvrSr35 while Sr35^L11E/L15E was expressed without a tag. The Sr50 resistosome was purified by a single-step affinity co-immunoprecipitation via the C-terminal Twin-Strep tag on Sr50^L11E/L15E, while AvrSr50 was expressed without a tag. The 2.5 mL of Twin-Strep eluate was concentrated and analysed by SEC as described above.

## Negative staining and TEM

Carbon film grids were glow discharged for negative staining of protein samples. The MLA13^K98E/K100E–AVR_A13-1 heterodimer, Sr35 resistosome and Sr50 resistosome samples were series-diluted in buffer B. Six microlitres of sample were applied to the grid and incubated for one minute before blotting off excess sample with filter paper. Six microlitres of one percent uranyl acetate were then applied to the grids and incubated for one minute before blotting off with filter paper.

Grids were then analysed by TEM.

## Cryo-EM sample preparation and data collection

Three microlitres of the purified MLA13$^{K98E/K100E}$–AVR$_{A13}$-1 sample were applied to an untreated graphene oxide-coated TEM grid, incubated on the grid for 10 s, blotted for 5 s and flash-frozen in liquid ethane using a Vitrobot Mark IV set to 90% humidity at 4 °C. Grids were stored under liquid nitrogen conditions until usage.

Cryo-EM data was acquired using a Titan Krios G3i electron microscope operated at 300 kV. Images were collected automatically using EPU on a Falcon III direct electron detector with a calibrated pixel size of 0.862 Å*px$^{-1}$. Target defocus values were set to −2.0 to −0.3 μm. Data was acquired using a total dose of 42 e$^{-}$*Å$^{-2}$ distributed among 42 frames, although the last three frames were excluded during data analysis.

## Image processing and model building

Image processing was performed using CryoSPARC. Movie stacks were first corrected for drift and beam-induction motion, and then used to determine defocus and other CTF-related values. Only high-quality micrographs with low drift metrics, low astigmatism, and good agreement between experimental and calculated CTFs were further processed. Putative particles were automatically picked based on an expected protein diameter between 8 and 12 nm, then extracted and subjected to reference-free 2D classification. 2D classes showing protein-like shapes were used for a template-based picking approach. Candidate particles were extracted again, subjected to reference-free 2D classification to exclude artefacts, and subsequent 3D classification to identify high-quality particles showing defined density for the effector, NBD, and LRR. This subset of particles was further refined using the non-uniform refinement strategy, yielding a map at a global resolution of 3.8 Å. DeepEMhancer was used to optimise the map for subsequent structure building. For further details, see Fig. EV2.

AlphaFold was used to predict a model for the CC-NBD-LRR domains of MLA13 from *H. vulgare* using the sequence Q8GSK4 from UniProt and two previously deposited structures in the PDBe, 5T1Y and 3QFL. The AlphaFold-predicted model was fitted into the map; however, the fold of the CC domain did not match the observed density adjacent to the LRR. Afterwards, Robetta was used to predict only this region, which gave outputs that more closely resembled the activated form of ZAR1 resistosome's CC domain. Robetta uses deep learning-based methods, RoseTTAFold and TrRosetta algorithms, and thus it may be influenced by existing models of the sequence to be predicted. For this reason, the ab initio option was chosen when running a second round of predictions in Robetta, and a template of the inactive ZAR1 CC domain from *A. thaliana* (PDB: 6J5W, Wang et al, 2019b) was included in the subsequent prediction run. The new model of the CC domain fitted the Coulomb potential map significantly better than the previous predicted models; thus, it was merged with the rest of the MLA13 model for refinement. Finally, the model containing AVR$_{A13}$-1-bound MLA13$^{K98E/K100E}$ was refined against the Coulomb potential map in iterations of *phenix.real_space_refine* and manual building in Coot. For further details and statistics, see Appendix Table S1. Molecular visualisation and analysis were done using UCSF ChimeraX.

## Cell death assays in barley protoplasts

Experiments were performed according to Saur et al (2019a) with the exception that plasmid DNA of all constructs was diluted to 500 ng/μL and transfection volumes were 15 μL, 10 μL and 10 μL for *pUBQ:luciferase*, *Mla*, and *AVR$_a$*, respectively (Saur et al, 2019b).

## Cell death assays in leaves of *N. benthamiana*

DNA of effector and receptor sequences were cloned as mentioned above into pGWB402SC and pGWB517, respectively. Transformation and preparation of *A. tumefaciens* suspensions was performed as mentioned above. Phenotype images were taken 72 h post infiltration while samples for western blot analysis were harvested 24 h post infiltration.

Western blotting of samples consisted of flash-freezing 100 mg of each sample and pulverising the tissue using a bead beater. The frozen leaf powder was resuspended in the aforementioned buffer A. The samples were centrifuged twice at 16,000 RCF before adding 4× Lämmli buffer supplemented with 5% mM β-mercaptoethanol and heating the sample to 95 °C for 5 min before cooling on ice. Ten microlitres of each sample were run on SDS-PAGE gels of indicated percentages before transferring to a PVDF membrane. The membranes were then blocked in TBS-T containing 5% milk for one hour at room temperature (RT). Membranes were washed three times for 5 min in TBS-T then incubated with anti-HA (1:2000) and anti-MYC (1:5000) in TBS-T with 5% BSA for one hour at RT. Membranes were washed in TBS-T for 3×10 min incubating with secondary anti-rabbit (1:2000) and anti-mouse (1:5000) in TBS-T with 5% milk for one hour at RT. Membranes were washed in TBS-T for 3×15 min before developing using SuperSignal West Femto substrate.

## Co-IP assays

Agroinfiltration of receptor and effector constructs into leaves of *N. benthamiana* were left to incubate for 20 h post infiltration. Two hundred milligrams of leaf tissue of each treatment was harvested, flash-frozen and pulverised using a bead beater. The tissue was thawed to 4 °C in 400 μL of the same buffer A as described above. The samples were centrifuged twice at 16,000 RCF. Forty microlitres of Strep-Tactin® XT Sepharose resin for each treatment was equilibrated in 1 mL of the above-described buffer B as and collected by centrifugation at 100 RCF for 1 min. The wash buffer supernatant was discarded before adding the 400 μL of sample supernatant to the resin. The sample-resin suspension was mixed by end-over-end rotation at 4 °C for 30 min. The resin was collected by centrifugation at 100 RCF for 1 min. and washed three times by resuspending in 1 mL of wash buffer. The final wash buffer supernatant was removed, after which 30 μL of wash buffer and 10 μL of 4× Lämmli buffer were supplemented with 5% mM β-mercaptoethanol. The samples were briefly vortexed and heated to 95 °C for five minutes before cooling on ice. The resin was collected by centrifugation at 100 RCF for 1 min when 15 μL of the supernatant was loaded onto a 12% SDS-PAGE gel. The input samples were treated the same as in the above section describing western blotting methods. Detection of all proteins were treated the same as in the above section describing western blotting methods.

## Data availability

The Coulomb potential map has been deposited in the EMDB under the accession code EMD-50863 (https://www.ebi.ac.uk/emdb/EMD-50863). Atomic coordinates have been deposited in the Protein Data Bank under the accession code 9FYC (https://www.rcsb.org/structure/9FYC). Other data used to generate tables and figures has been provided as source data with this publication.

The source data of this paper are collected in the following database record: biostudies:S-SCDT-10_1038-S44318-025-00373-9.

## Peer review information

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

## Acknowledgements

The authors thank the greenhouse team at MPIPZ for their expertise in providing high-quality *N. benthamiana* plants. The authors thank Neysan Donnelly and Jane Parker for critical comments on an early version of this manuscript. The authors thank Arthur Macha, Petra Koechner, Sabine Haigis, Elke Logemann, Milena Malisic, Florian Kuemmel, Li Liu, Wen Song and Nitika Mukhi for their intellectual and experimental contributions. This work was funded by the Max-Planck-Gesellschaft (PS-L), the Deutsche Forschungsgemeinschaft (DFG, German Research Foundation) in the Collaborative Research Centre Grant (SFB-1403— 414786233 B08 to PS-L, IMLS and JC), Germany's Excellence Strategy CEPLAS (EXC-2048/1, project 390686111 to PS-L and IMLS), the Ministry of Culture and Science of the State of North Rhine-Westphalia (iHEAD to PS-L and EB) and DFG Emmy Noether Programme (SA 4093/1-1 to IMLS). The authors acknowledge access to the cryo-EM infrastructure of StruBiTEM (Cologne, funded by DFG Grant INST 216/949-1 FUGG), and to the computing infrastructure of CHEOPS (Cologne, funded by DFG Grant INST 216/512/1 FUGG).

## Author contributions

**Aaron W Lawson**: Conceptualisation; Investigation; Data analysis; Writing; Editing. **Andrea Flores-Ibarra**: Structural model building. **Yu Cao**: Investigation. **Chunpeng An**: Investigation. **Ulla Neumann**: Electron microscopy screening. **Monika Gunkel**: Electron microscopy screening; data analysis. **Isabel M L Saur**: Investigation. **Jijie Chai**: Conceptualisation; Project administration. **Elmar Behrmann**: Conceptualisation; Project administration; Data analysis; Structural model building; Writing; Editing. **Paul Schulze-Lefert**: Conceptualisation; Project administration; Writing; Editing.

Source data underlying figure panels in this paper may have individual authorship assigned. Where available, figure panel/source data authorship is listed in the following database record: biostudies:S-SCDT-10_1038-S44318-025-00373-9.

## Funding

## Disclosure and competing interests statement

The authors declare no competing interests.

# Expanded View Figures

**Figure EV1. Co-expression of MLA13–AVR$_{A13}$-1 without an N-terminal GST tag on MLA13 or with various MLA13 substitutions consistently elute at a volume indicating a low-molecular-weight complex.**

Elution volumes of the Sr35 resistosome, Sr50 resistosome and MLA13$^{K98E/K100E}$-AVR$_{A13}$-1 heterodimer are shown with a dotted line when purified using the same method (Appendix Figs. S3, S4 and main Fig. 1A, respectively). (A) MLA13$^{K98E/K100E}$-2S-HA expressed and purified alone. (B) MLA13$^{K98E/K100E/D502V}$-2S-HA expressed and purified alone. (C) MLA13$^{K98E/K100E}$-GST expressed with AVR$_{A13}$-2S-HA. (D) MLA13$^{K98E/K100E}$ expressed with AVR$_{A13}$-2S-HA. (E) MLA13$^{L11E/L15E}$ expressed with AVR$_{A13}$-2S-HA. All samples were purified from 100 g of leaf tissue and with a single-step affinity purification via the Twin-Strep-tag® (2S-HA), followed by SEC. Fractions from the SEC profiles are numbered and presented on the accompanying CBB-stained, 10 or 12% SDS-PAGE gels. Source data are available online for this figure.

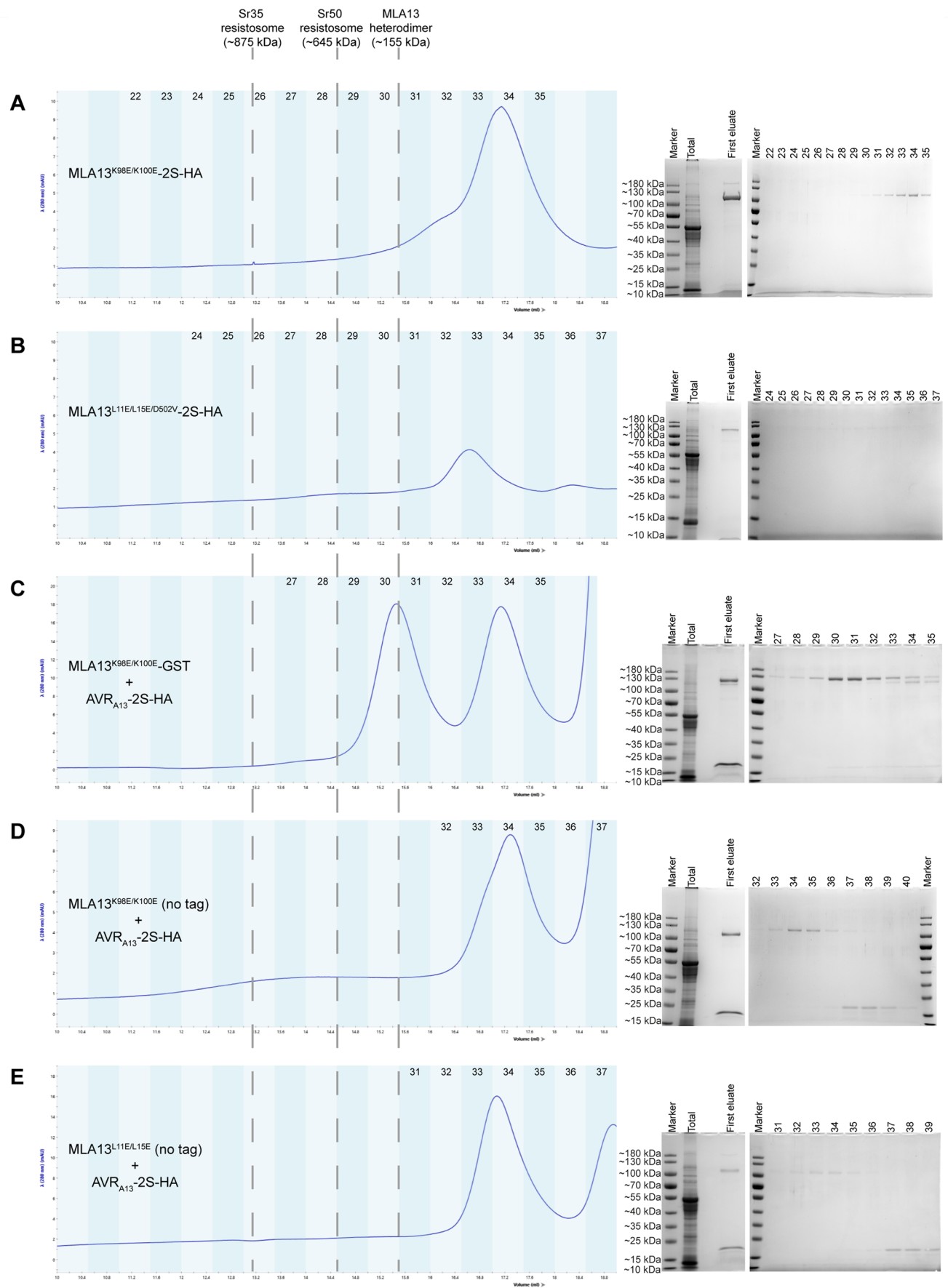

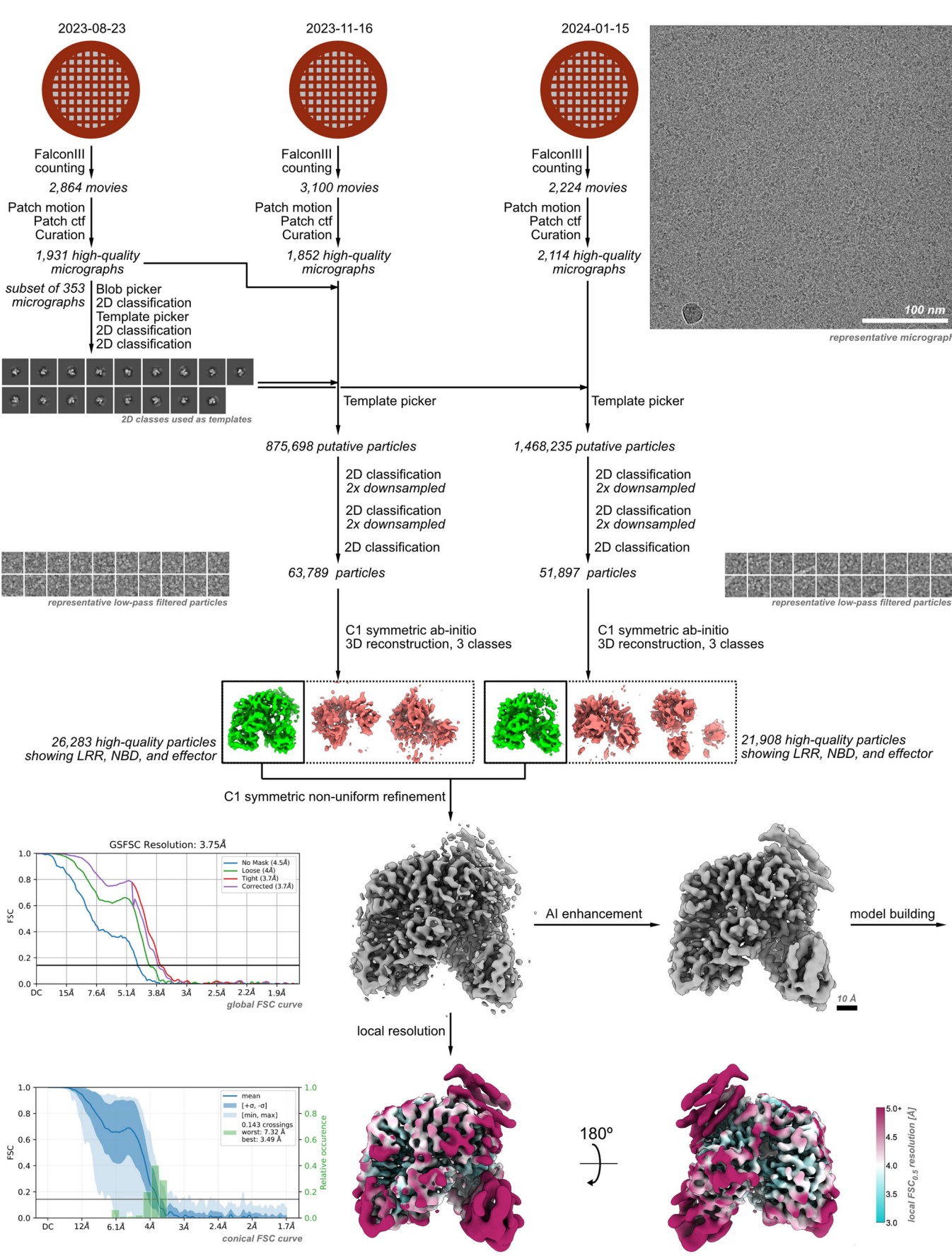

**Figure EV2.  Workflow of cryo-EM data acquisition and analysis of the MLA13$^{K98E/K100E}$-AVR$_{A13}$-1 heterodimer.**

A total of three datasets were collected on a 300 kV cryo-electron microscope. For each dataset, movies were selected for low per-frame drift rates, good CTF scores, and low astigmatism. Particles were first picked using a blob picker, and then subjected to unsupervised 2D classification. Representative classes showing protein-like structures were used for a template picker. Putative detected particles were curated using unsupervised 2D classification, selecting for particles with protein-like density and resolutions better than 10 Å. The selected particles were further curated using ab initio reconstruction, sorting them into three distinct populations. From these, all particles contributing to a structure showing clear density for the LRR, NBD and effector (shown in green and highlighted by a thicker box outline) were combined and refined in 3D using a non-uniform refinement algorithm, resulting in a map with a global resolution of 3.8 Å. Before model building, the map was further sharpened using DeepEMhancer.

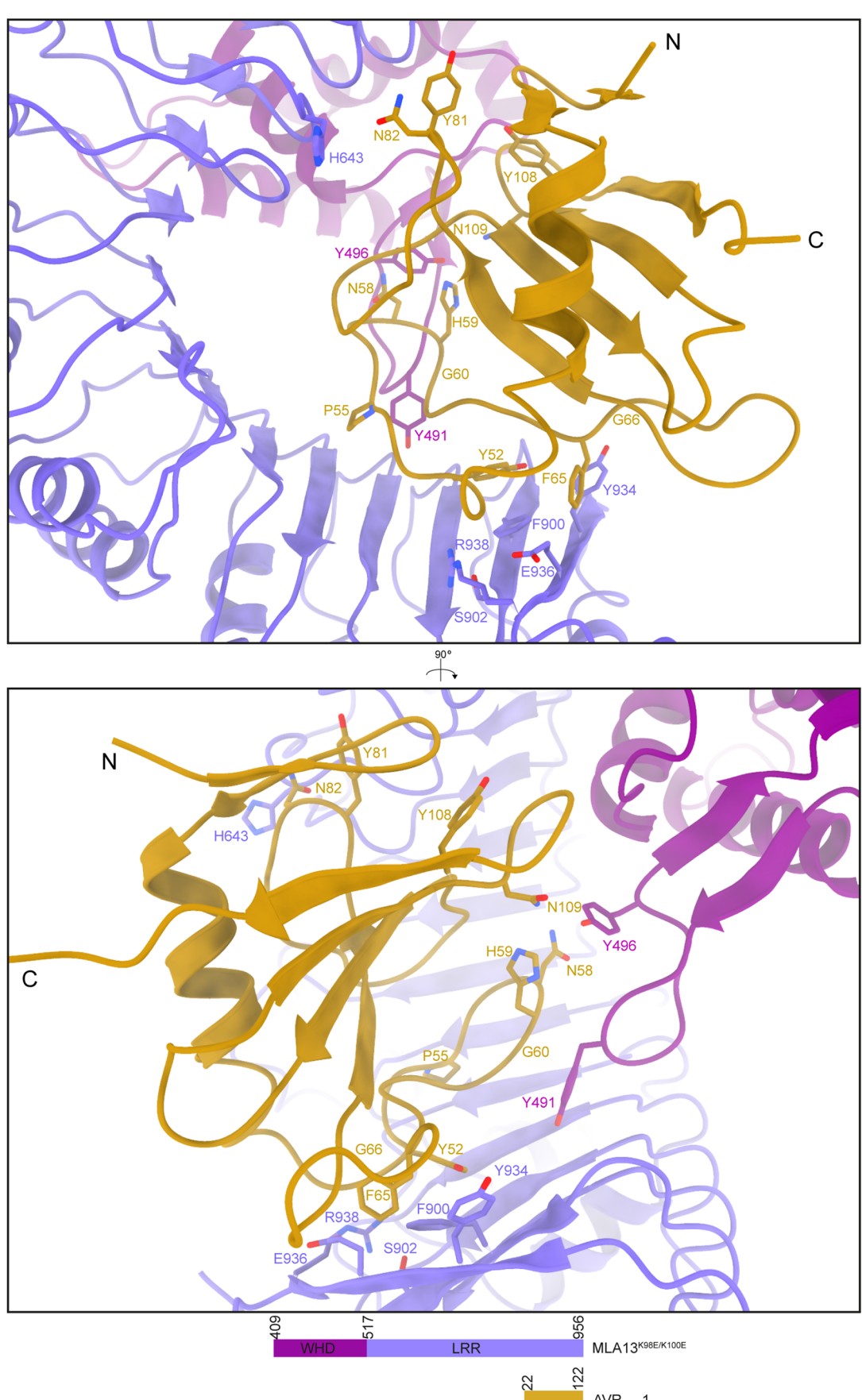

◀ **Figure EV3. The MLA13^{K98E/K100E}-AVR_{A13}-1 interface from two different angles.**

Atomic model showing residues predicted to contribute to the interface and/or experimentally tested for loss of MLA13-mediated cell death.

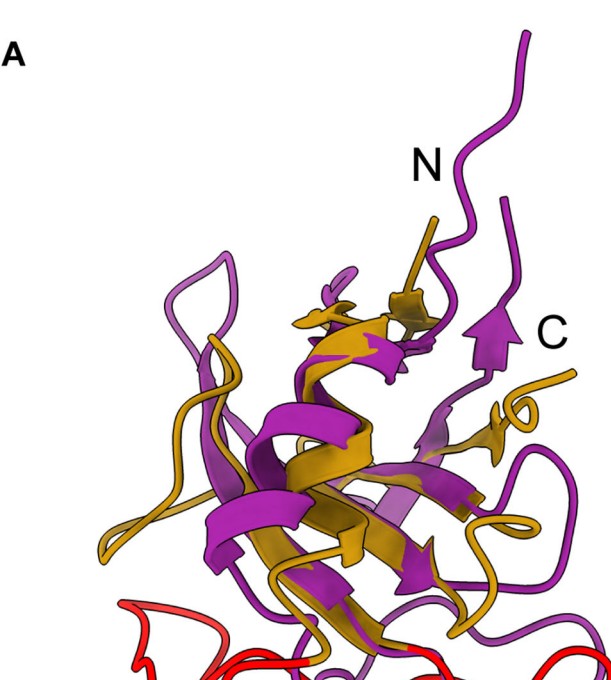

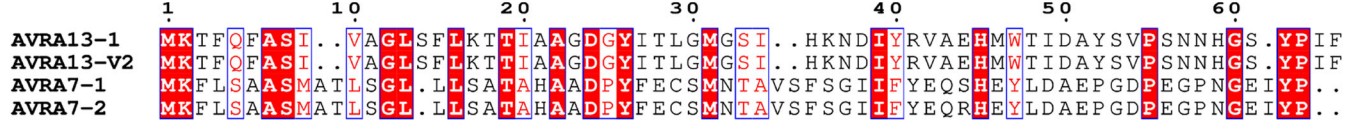

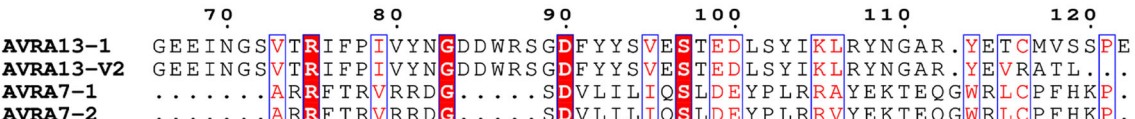

**Figure EV4.  Structural and sequence alignments of AVR<sub>A13</sub> and AVR<sub>A7</sub> variants.**

(A) Structural alignment of AVR$_{A13}$-1 (dark goldenrod colour) and crystal structure of AVR$_{A7}$-1 (burgundy colour; PDB: 8OXL). The basal loops of AVR$_{A13}$-1 are coloured in red. (B) Sequence alignment of AVR$_{A13}$ and AVR$_{A7}$ variants. Alignment performed using MUSCLE and visualised using ESPript 3.0.

