## [Peer Review File · The EMBO Journal]

The barley MLA13-AVRA13 heterodimer reveals principles for immunoreceptor recognition of RNase-like powdery mildew effectors

Aaron Lawson, Andrea Flores-Ibarra, Yu Cao, Chunpeng An, Ulla Neumann, Monika Gunkel, Isabel Saur, Jijie Chai, Elmar Behrmann, and Paul Schulze-Lefert

Corresponding author(s): Paul Schulze-Lefert (schlef@mpipz.mpg.de) , Jijie Chai (chajijie@westlake.edu.cn), Elmar Behrmann (elmar.behrmann@uni-koeln.de)

Review Timeline:

Submission Date:	3rd Aug 24
Editorial Decision:	26th Aug 24
Revision Received:	5th Dec 24
Editorial Decision:	14th Jan 25
Revision Received:	16th Jan 25
Accepted:	20th Jan 25

Editor: William Teale

Transaction Report:

Dear Paul,

Thank you again for the submission of your manuscript entitled "The barley MLA13-AVRA13 heterodimer reveals principles for immunoreceptor recognition of RNase-like powdery mildew effectors" and for your patience during the review process. We have now received three reports from the referees, which I copy below.

As you can see from their comments, all referees considered the work you present to be timely and well conducted; furthermore, the research question you ask is well-defined and of interest to the readership of The EMBO Journal. That said, referees #2 and #3 both raise the question of the biological nature of the MLA13/Avr13A-1 heterodimer. Clarification in this direction will require your attention before your manuscript can be published in The EMBO Journal.

Based on the overall interest expressed in the reports, though, I would like to invite you to address the comments of all referees in a revised version of the manuscript. I should add that it is The EMBO Journal policy to allow only a single major round of revision and that it is therefore important to resolve the main concerns at this stage. I believe the concerns of the referees are reasonable and addressable, but please contact me if you have any questions, need further input on the referee comments or if you anticipate any problems in addressing any of their points. If you would like to discuss these reports with me, I would be happy to talk via Zoom once you have had a chance to reflect on them fully. I would be especially interested in whether you can add any data that supports the MLA13/Avr13A-1 heterodimer being an intermediate step in resistosome assembly. We could discuss what data are both feasible to collect and likely to satisfy the reviewers. Please, follow the instructions below when preparing your manuscript for resubmission.

I would also like to point out that as a matter of policy, competing manuscripts published during this period will not be taken into consideration in our assessment of the novelty presented by your study ("scooping" protection). We have extended this 'scooping protection policy' beyond the usual 3 month revision timeline to cover the period required for a full revision to address the essential experimental issues. Please contact me if you see a paper with related content published elsewhere to discuss the appropriate course of action.

Again, please contact me at any time during revision if you need any help or have further questions.

Thank you very much again for the opportunity to consider your work for publication. I look forward to your revision.

Best regards,

William

William Teale, Ph.D.
Editor
The EMBO Journal

When submitting your revised manuscript, please carefully review the instructions below and include the following items:

- 1) a .docx formatted version of the manuscript text (including legends for main figures, EV figures and tables). Please make sure that the changes are highlighted to be clearly visible.
- 2) individual production quality figure files as .eps, .tif, .jpg (one file per figure).
- 3) a .docx formatted letter INCLUDING the reviewers' reports and your detailed point-by-point response to their comments. As part of the EMBO Press transparent editorial process, the point-by-point response is part of the Review Process File (RPF), which will be published alongside your paper.
- 4) a complete author checklist, which you can download from our author guidelines ([https://wol-prod-cdn.literatumonline.com/pb-assets/embo-site/Author Checklist%20-%20EMBO%20J-1561436015657.xlsx](https://wol-prod-cdn.literatumonline.com/pb-assets/embo-site/Author%20Checklist%20-%20EMBO%20J-1561436015657.xlsx)). Please insert information in the checklist that is also reflected in the manuscript. The completed author checklist will also be part of the RPF.
- 5) Please note that all corresponding authors are required to supply an ORCID ID for their name upon submission of a revised manuscript.

6) We require a 'Data Availability' section after the Materials and Methods. Before submitting your revision, primary datasets produced in this study need to be deposited in an appropriate public database, and the accession numbers and database listed under 'Data Availability'. Please remember to provide a reviewer password if the datasets are not yet public (see <https://www.embopress.org/page/journal/14602075/authorguide#datadeposition>). If no data deposition in external databases is needed for this paper, please then state in this section: This study includes no data deposited in external repositories. Note that the Data Availability Section is restricted to new primary data that are part of this study.

Note - All links should resolve to a page where the data can be accessed.

8) For data quantification: please specify the name of the statistical test used to generate error bars and P values, the number (n) of independent experiments (specify technical or biological replicates) underlying each data point and the test used to calculate p-values in each figure legend. The figure legends should contain a basic description of n, P and the test applied. Graphs must include a description of the bars and the error bars (s.d., s.e.m.).

9) We would also encourage you to include the source data for figure panels that show essential data. Numerical data can be provided as individual .xls or .csv files (including a tab describing the data). For 'blots' or microscopy, uncropped images should be submitted (using a zip archive or a single pdf per main figure if multiple images need to be supplied for one panel). Additional information on source data and instruction on how to label the files are available at .

10) We replaced Supplementary Information with Expanded View (EV) Figures and Tables that are collapsible/expandable online (see examples in <https://www.embopress.org/doi/10.15252/emboj.201695874>). A maximum of 5 EV Figures can be typeset. EV Figures should be cited as 'Figure EV1, Figure EV2' etc. in the text and their respective legends should be included in the main text after the legends of regular figures.

12) Our journal encourages inclusion of *data citations in the reference list* to directly cite datasets that were re-used and obtained from public databases. Data citations in the article text are distinct from normal bibliographical citations and should directly link to the database records from which the data can be accessed. In the main text, data citations are formatted as follows: "Data ref: Smith et al, 2001" or "Data ref: NCBI Sequence Read Archive PRJNA342805, 2017". In the Reference list, data citations must be labeled with "[DATASET]". A data reference must provide the database name, accession number/identifiers and a resolvable link to the landing page from which the data can be accessed at the end of the reference. Further instructions are available at .

13) In order to increase the reproducibility and reach of your work, The EMBO Journal includes a table of reagents that were used in the study. Please provide this along with your revisions.

We realize that it is difficult to revise to a specific deadline. In the interest of protecting the conceptual advance provided by the work, we recommend a revision within 3 months (24th Nov 2024). Please discuss the revision progress ahead of this time with the editor if you require more time to complete the revisions. Use the link below to submit your revision:

Referee #1:

The paper by Lawson et al. demonstrates the potential of using structural information to guide NLR engineering for new recognition capabilities. While the study is impressive, several limitations need to be addressed. The underlying mechanisms involved in the engineering process are not thoroughly explored, and the assertion that the MLA13/Avr13A-1 heterodimer represents an intermediate state lacks sufficient evidence. Additionally, several figures in the manuscript require significant improvements in labelling and clarity, and further experiments are necessary to better support the conclusions. Although the study provides valuable insights, addressing these issues would greatly enhance the paper's conclusions and overall impact. The conformation of the receptor bound to the effector turned out to be unexpected, as this reveals them as a heterodimer complex of NLR and effector rather than an assembled resistosome. With this 3D-structure, the authors report this as an intermediate state/CNL output that is in equilibrium with resistosome formation. For this study, the authors used a triple mutant (MLA13 K98E/K100E/D502V). As such, the structure is similar to available ZAR monomers (PDB codes 6J5W, 6J5V) with differences in the details of NB-domain orientation. In addition to this unexpected conformation, the structure also reveals the details of molecular recognition between the cognate effector and the NLR in this conformation, and also provides a structural rationale for how host proteins containing the same effector fold are excluded from the recognition and avoid mis-activation of immunity. Overall, the paper provides new information in the structural understanding of NLR action. But the authors need to explain more details on the nature of MLA13 triple mutant and its implication for the accumulation of heterodimer and absence of resistosomes in this study. This commentary can be included in the introduction part and in discussion part.

Detailed comments are provided below.

Major:

1. Line 133 and Extended Data Fig. 3

a) Extended Data Fig. 3a showed that GST-MLA13 didn't trigger HR with the presence or absence of its cognate Avr.

Introducing the autoactive D502V mutation into GST-MLA13 also fails to induce HR. These results raise questions of whether

the N-terminal GST tag is interfering with MLA13 function. If so, how does this impact the validity of the reported GST-MLA13/AvrA13-1 structure?

2. Extended Data Fig. 6: This figure presents conflicting data that require clarification:

The left panel demonstrates that MLA13L11E/L15E, in comparison to Sr35 L11E/L15E, does not oligomerize in the presence of its cognate Avr. However, given that MLA13L11E/L15E forms stable heterocomplexes with AvrA13-1, we would expect to observe a size increase of MLA13 due to effector binding. AvrA13-1-GFP is approximately 40 kDa (as shown in the right panel), which should result in a noticeable size shift of MLA13 in BN-PAGE upon binding to it. However, this expected size shift is not evident (left panel, last three lanes). These data thus support the authors' conclusion in one way but conflict in another. The authors should provide a detailed explanation for:

a) Why is the expected size shift not observed upon AvrA13-1-GFP binding? Please provide the immunoblot results using an anti-GFP antibody, showing where the AvrA13-1-GFP signal is in BN-PAGE.

b) How this result reconciles with the claim of stable heterocomplex formation.

c) Or whether there are any technical limitations in the assay that might explain this unexpected result.

3. Lines 222-224: The authors' interpretation of the MLA13/AvrA13-1 heterodimer complex as an "intermediate state after effector binding-induced release of ADP but before ATP binding-induced oligomerisation" requires further validation. Unlike Zar1, MLA13/AvrA13-1 complex was expressed in the *N. benthamiana* system that should provide all necessary components (including ATP) for full NLR activation. It's unexpected that a functional NLR is stuck at an intermediate state but not the final conformation. The authors should consider alternative explanations, such as the possibility that the observed heterodimers are fragments of higher-order oligomers that broke down during purification.

To conclusively demonstrate the *in vivo* state of the MLA13/AvrA13-1 complex and avoid potential artifacts from tags and purification procedures, the authors should conduct additional experiments:

a) Co-expressing functional MLA13/AvrA13-1 combinations (capable of triggering cell death) in *N. benthamiana* and analyze these samples using BN-PAGE without immunoprecipitation. This would allow detecting MLA13/AvrA13-1 in their native state. Co-infiltration with LaCl₃ could be considered to suppress cell death to enhance protein accumulation.

b) Checking whether adding ATP or dATP to the purified MLA13/AvrA13-1 heterodimers triggers oligomerization.

c) Extracting the nucleotide binding to the heterodimers and check its identity by HPLC.

4. Fig.3 and 4 Please include the corresponding interaction analyses, such as co-IP, split-luciferase assay, or BiFC, for all the residue substitution analyses shown in Figures 3 and 4. These additional tests are essential to verify that the observed loss of cell death resulting from residue substitutions is indeed due to a disruption in interaction, rather than other potential factors, such as inhibiting conformational changes.

5. The engineering part: The authors describe how they identified the residues MLA7(L902) and MLA13(S902) as potentially responsible for the differential recognition between MLA7 and MLA13. They note that the residue-switched variant MLA7L902S can confer new recognition of Avr13A-1. But the mechanism such as how it works is not well explained. When examining Fig. 4, I noticed that MLA13(S902) actually contributes little to its recognition of Avr13A-1. Specifically, the mutant variant MLA13S902A still triggers strong cell death when co-expressed with Avr13A-1 (Figure 4C). This observation raises questions about the mechanism by which the MLA7L902S swap functions:

a) Does Leu902 in MLA7 cause steric clashes with Avr13A-1, thereby preventing interaction? And the substitution to a smaller residue like Serine remove this clash and allow recognition?

b) To test this hypothesis, the authors could mutate MLA7(L902) to other small residues with short side chains and assess whether these variants can recognize Avr13A-1. Additionally, they could explore the effects of substituting MLA13(S902) with Leu or other large side chain residues, or with other small side chain residues.

c) Finally, it is essential to test the interactions of all these variants, either by co-IP, split-luciferase assay, or by BiFC.

Other comments:

1. Extended Data Fig. 3 Despite the significant difference in size between myc and GST tags, myc-MLA13 variants in Extended Data Fig. 3b appear to have similar molecular weights to GST-MLA13 shown in Extended Data Fig. 1. Could the author explain this and give a bit more detail of how they performed these experiments? Improved descriptions in the legends are necessary for better clarity and understanding.

2. Fig1a The figure is confusing and requires improvement in labelling and clarity. The authors state that the inserted SDS-PAGE gel represents "fractions eluted along the black line". However, the placement of the 875 kDa arrow at the end of the black line suggests that the samples are loaded in reverse order - from large elution volume to small. This contradiction needs clarification.

3. Fig. 2C, D Please provide the prediction scores (pLDDT and PAE plots, pTM and ipTM) for AF3 models.

4. Fig. 3e Data here is expected to provide evidence that MLA13 was expressed among all the panels in Fig. 3C, especially the combinations losing cell death. The current figure is meaningless. Same issue with Fig. 4e.

5. Fig3a and 4a The separation of interacting residues between the receptor and effector in Fig. 3a (effector side) and Fig. 4a (NLR side) makes it difficult to comprehend the complete interaction interface. While this approach keeps each figure tidy, it hinders understanding of which NLR residues interact with specific effector residues. An additional merged figure is recommended, showing both NLR and effector residues involved in the interaction.

In addition:

- What is the net molecular weight of the heterodimer used in this structural study in kDa? This has to be included somewhere in the paper as this is a cryoEM study of intermediate size protein complex.

- Where does K98 and K100 come in the 3D structure of monomeric ZAR1? It is in the CC domain, but the exact location of these residues and their interactions must be discussed. For example, are these residues lying in the allosteric path that leads to

fold switch of CC domain N-terminal alpha1 helix? Are they interacting with the alpha1 helix in the monomeric ZAR1? These details will clarify the questions on whether the mutation K98E, K100E impacts the conformational transition required to release the alpha1 helix after effector recognition and assemble into resistosome in an autoactive MLA D502V mutant.

- In Line 137, the authors suggest that analogous substitution in NRG1 retains oligomerisation. Although ZAR1, MLA13 and NRG1 are CNLs, we know that NRG1 forms a phylogenetically different class of RPW8-NLRs, so their resistosome assembly might be different from that of ZAR1 or MLA13. It is possible that they form similar resisting state structure but have a different resistosome structure.
- In Line 146-147, the authors suggest that SEC revealed a heterocomplex with size smaller than multimeric MLA13. Since there is cell death in this triple mutant, it is possible that the multimers are just at very low concentration. Would it be a good idea in that case, to pull-down the MLA13 effector complex first with MLA's flag-tag and screen in negative stain before going for second pull-down from effector side? Such negative stain images immediately after first pull-down from crude extract without SEC may reveal a lower number of existing resistosomes, though not homogeneous.
- In Figure 2, the legend describes as 'electron density map'. The authors know very well that the map from cryo-EM reconstruction is a 'coulomb potential map' and not electron density map as in X-ray crystallography. This typing mistake has to be corrected.
- In Line 271 and 275, MLA13 residue 934 is given as F and Y respectively. The atomic model displayed in the Figure 4A, shows it as Y934. This typing mistake has to be corrected in Line 271.
- Here is a general observation in extended figure 6, left panel. The last but one lane, where MLA13 alone is loaded, runs in the BN-PAGE near 242kDa, but its molecular weight is around 107kDa for single chain. Such migration of the apoprotein in resting state is an indication for a dimeric state in resting state as observed for NRC2. Resting NRC2 also migrates near this location in BN-PAGE since they are dimers. The authors can track down this if they are interested in resolving the resting state of MLA.
- In the Table 1 of extended data, there is no description for 'Energy filter' is given nor mentioned in the cryo-EM imaging section. Small protein complexes normally benefit from the use of energy filter during imaging. Was there any particular reason why it was not used in this imaging experiment? If there is any advantage, information along this will be useful for all others working on this range of molecular size.
- Finally, the heterocomplex structure describes how this NLR is engaging its different parts to recognize its cognate effector. Does the authors think, the effector position and the recognition areas of NLR change when this heterocomplex at non-canonical conformation moves to oligomeric, resistosome conformation. Can the initial recognition mode of the effector and its final position in assembled resistosome be expected to vary? All views of the authors along this will be useful in the discussion on engineering perspectives.

Referee #2:

In this study, the authors show the structural basis of MLA13 recognition of AVRA13 and manage to modify the specificity of the NLR which is very impressive. The description of NLR/effector recognition is well supported and very convincing.

I have one major point regarding the biological relevance of the MLA13 K98/K100 mutant used in this study.

Is this MLA13 heterodimer active and does it reflect the function of the WT protein? Does the WT protein exist in vivo as a heterodimer similar to the one seen here? Do we see higher order oligomers of MLA13 WT in planta following AVR13 recognition? The claim that this dimeric state represents an intermediate step of MLA13 activation is weakly supported by data. The authors should provide information on the function of MLA13 K98E/K100E compared to WT. Ideally, the authors should show that the WT protein form higher order oligomers following AVRA13 recognition, that the K98/K100 D502V forms similar oligomers and determine the structure of the K98/K100 D502V mutant oligomer. Is MLA13 WT also sensitive to calcium channel inhibitors? At least, the author should provide data on MLA13 K98/100 D502V oligomerization status.

Please justify the use of the L11/15 D502V mutant for the determination of oligomerization in blue native gels. Why did the authors use the K98/100 mutant for the structural study instead of the L11/15 mutant? This mutant makes higher order oligomers similar to active resistosomes. It is unclear in extended figure 6 if the D502V mutation is combined with the L11/15 mutations. Please specify in the caption.

Please specify in the figures when MLA13 K98/100 mutants are used. This information is essential.

Where are residues L98/100 used for purifying MLA13? How do they affect MLA13 function? Please provide a view of the structure with these residues highlighted. Also in figure 1, please indicate the number of amino-acids comprised in the different domains of MLA13 in the schematic below the structure.

Another major point. How does the purification process affect the complex retrieved? Do you observe the same SEC profile by

pulling down MLA13? Does AVRA13 interact with oligomeric forms of MLA13 such as the one seen in extended figure 6? If not, maybe the purification process enriches heterodimers and not full resistosomes?

Minor point:

Line 133, figure numbering is off, extended figure 3 is cited before extended figure 1.

Referee #3:

This manuscript presents the cryo-EM structure of barley CNL immune receptor MLA13 in complex with its cognate effector AVRA13-1 as a heterodimer. In contrast to previously reported effector activated CNL pentamers, they propose MLA13-AVRA13-1 heterodimer as an intermediate state. They validated the importance of the interactions between MLA13-AVRA13-1 for function. They further engineered a single amino acid substitution in MLA7 that enables expanded effector detection of AVRA13-1 and the virulent variant AVRA13-V2. This study significantly advanced the understanding on barley MLA receptors and their recognition of effectors. The manuscript is well written. I have a few comments for the authors to improve the manuscript.

1, In extended data Fig. 6, BN-PAGE showed the putative oligomer of MLA13L11E/L15E/D502V. Can the authors also purify MLA13L11E/L15E/D502V and do negative stain of it as they did for Sr35 resistosome?

2, For Nb cell death images, can the authors please make a mock leaf showing the infiltration settings for readers to better understand?

3, In extended data Fig. 1, please run all samples on one gel.

We thank the reviewers for their interest in our work and their constructive
suggestions, which greatly contributed to the improvement of our manuscript.

During the revision of our manuscript, a preprint from another laboratory has
been posted that reports the results of *in vitro* shuffling of barley *Mla* genes
(<https://www.biorxiv.org/content/10.1101/2024.10.27.619561v1>). Remarkably, these
authors show that after recombination of *Mla7* and *Mla13* genes *in vitro* and
subsequent screening of the resulting library, substitution of the exact same amino
acid substitution in MLA7 that we had engineered based on our structural knowledge
facilitated interaction with AVR_{A13-1} in yeast and AVR_{A13-1}-dependent and receptor-
mediated cell death *in planta*. Together, this strongly supports the physiological
relevance of our structure-based study for understanding effector-triggered MLA-
mediated cell death activation.

Referee #1:

The paper by Lawson et al. demonstrates the potential of using structural information
to guide NLR engineering for new recognition capabilities. While the study is
impressive, several limitations need to be addressed. The underlying mechanisms
involved in the engineering process are not thoroughly explored, and the assertion
that the MLA13/Avr13A-1 heterodimer represents an intermediate state lacks
sufficient evidence.

We appreciate these comments. Due to the unexpected heterodimeric conformation
of the MLA13-AVR_{A13-1} heterocomplex, we limited our conclusions in the original
manuscript to cautious conjecture, using 'may' or 'might.' Our only conclusions
regarding the MLA13^{K98E/K100E}-AVR_{A13-1} heterodimer and its activity are limited to
structure-guided substitutions at the effector-binding interface that result in a loss of
cell death and a common structural principle for RNase-like effector binding to MLAs.
This is also reflected in the title of our manuscript.

Additionally, Extended Data Figure 7 in the original version of the manuscript
presents SEC profiles of samples with alternative substitutions (i.e., MLA13^{L11E/L15E})
and MLA13 without the N-terminal GST tag (now reformatted as Figure EV1). These
samples consistently elute at a volume similar to the GST-MLA13^{K98E/K100E}-AVR_{A13-1}
heterodimer resolved by cryo-EM. This suggests that neither the N-terminal GST tag
on MLA13 nor the MLA13^{K98E/K100E} substitutions disrupt the formation of high-order
complexes, such as a hypothetical pentameric resistosome. This is detailed in line
181 of the original manuscript.

Additionally, several figures in the manuscript require significant improvements in
labelling and clarity, and further experiments are necessary to better support the
conclusions. Although the study provides valuable insights, addressing these issues
would greatly enhance the paper's conclusions and overall impact. The conformation
of the receptor bound to the effector turned out to be unexpected, as this reveals
them as a heterodimer complex of NLR and effector rather than an assembled
resistosome. With this 3D-structure, the authors report this as an intermediate
state/CNL output that is in equilibrium with resistosome formation.

Thank you for the comment. As mentioned in the first paragraph of the discussion of
the original manuscript, our rationale for the MLA13^{K98E/K100E}-AVR_{A13-1} heterodimer
as a possible intermediate state is based on the finding that the receptor

conformation is reminiscent of the nucleotide-free, ligand-bound intermediate
complex of ZAR1–RKS1 bound to PBL2^{UMP} (PDB: 6J5V). We conjecture that the
equilibrium between heterodimeric and resistosome conformations may be
differential among different CNLs (i.e. MLA13 versus Sr35). This text section has
been modified and highlighted yellow to emphasise our use of conjecture. We have
also removed the speculative sentence stating that “the equilibrium between
heterodimeric and pentameric resistosomes may be differentially regulated among
sensor CNLs”.

We cannot exclude that a hypothetical MLA13 resistosome disassembles into
receptor-effector heterodimers upon cell lysis and protein purification or is
inaccessible with our purification method. Regardless of whether the MLA13–
AVR_{A13}-1 heterodimer results from disassembly of a hypothetical MLA13
resistosome or is a stable ligand-bound intermediate state receptor, the established
receptor-effector interface is necessary for MLA13-mediated cell death. Accordingly,
in the revised discussion we now provide three possible explanations for the
MLA13^{K98E/K100E}–AVR_{A13}-1 heterodimeric conformation. “First, a hypothetical MLA13
resistosome may disassemble into receptor-effector heterodimers upon cell lysis and
protein purification or is otherwise inaccessible to our purification method. Second,
the concentration of a hypothetical MLA13 resistosome is too low compared to the
heterodimers to be isolated by our purification protocol, but still sufficient for
receptor-mediated cell death triggered by the effector. Third, the stable
MLA13^{K98E/K100E}–AVR_{A13}-1 heterodimer represents a non-pentameric, novel
immunostimulatory output. Of note, regardless of whether the MLA13^{K98E/K100E}–
AVR_{A13}-1 heterodimer results from disassembly of a hypothetical MLA13
resistosome or is a ligand-bound intermediate state receptor with immunostimulatory
activity, the observed receptor-effector interface is necessary for MLA13-mediated
cell death.”

For this study, the authors used a triple mutant (MLA13 K98E/K100E/D502V.

Please note that we did not purify and resolve the triple mutant
(MLA13^{K98E/K100E/D502V}; autoactive) as the reviewer stated above. We resolved the
double mutant MLA13^{K98E/K100E} (line 135 of the original manuscript). We have
described this more clearly in the revised text (highlighted yellow) and in Figure 1 of
the revised manuscript.

As such, the structure is similar to available ZAR monomers (PDB codes 6J5W,
6J5V) with differences in the details of NB-domain orientation. In addition to this
unexpected conformation, the structure also reveals the details of molecular
recognition between the cognate effector and the NLR in this conformation, and also
provides a structural rationale for how host proteins containing the same effector fold
are excluded from the recognition and avoid mis-activation of immunity. Overall, the
paper provides new information in the structural understanding of NLR action.
But the authors need to explain more details on the nature of MLA13 triple mutant
and its implication for the accumulation of heterodimer and absence of resistosomes
in this study.

Please note that we did not purify and resolve the triple mutant
(MLA13^{K98E/K100E/D502V}; autoactive) as the reviewer stated above. We resolved the

double mutant MLA13^{K98E/K100E} (line 135 of the original manuscript). We have
described this more clearly in the revised text (highlighted yellow) and in Figure 1 of
the revised manuscript. In addition, Extended Data Fig. 7 of the original manuscript
provides SEC profiles of samples with alternative substitutions (i.e. MLA13^{L11E/L15E}
and/or MLA13 without the N-terminal GST tag). These samples consistently elute at
a volume similar to that of the cryo-EM-resolved GST-MLA13^{K98E/K100E}-AVR_{A13-1}
heterodimer or later in the profile, suggesting that the N-terminal GST tag on MLA13
or the MLA^{K98E/K100E} does not disrupt the formation of a hypothetical oligomeric
receptor complex such as a pentameric resistosome. This is explained in line 181 of
the original version of the manuscript. Extended Data Fig. 7 of the original version of
the manuscript was reformatted as Figure EV1.

As mentioned above, we cannot exclude that a hypothetical MLA13
resistosome disassembles into receptor-effector heterodimers upon cell lysis and
protein purification or is inaccessible to our purification method. Regardless of
whether the MLA13-AVR_{A13-1} heterodimer results from disassembly of a
hypothetical MLA13 resistosome or is a stable ligand-bound intermediate state
receptor, the established receptor-effector interface is necessary for MLA13-
mediated cell death. We have added this to the second paragraph of the discussion.

Detailed comments are provided below.

Major:

1. Line 133 and Extended Data Fig. 3

a) Extended Data Fig. 3a showed that GST-MLA13 didn't trigger HR with the
presence or absence of its cognate Avr. Introducing the autoactive D502V mutation
into GST-MLA13 also fails to induce HR. These results raise questions of whether
the N-terminal GST tag is interfering with MLA13 function. If so, how does this
impact the validity of the reported GST-MLA13/AvrA13-1 structure?

Thank you for the comment. Extended Data Fig. 7c,d of the original manuscript show
that when AVR_{A13-1} is co-expressed and purified with either MLA13^{K98E/K100E} or
MLA13^{L11E/L15E} (both receptors without epitope tags; samples purified *via* C-terminal
Twin-Strep-tag on AVR_{A13-1}), both samples elute at a later elution volume (profile in
red; ~17.5 mL) than that of the resolved heterodimer (~15.5 mL), suggesting that
even when the receptor is expressed and purified without a tag it does not form an
oligomeric complex. This is explained in line 181 of the original version of the
manuscript. Extended Data Fig. 7 of the original version of the manuscript was
reformatted as Figure EV1.

2. Extended Data Fig. 6: This figure presents conflicting data that require
clarification:

The left panel demonstrates that MLA13L11E/L15E, in comparison to Sr35
L11E/L15E, does not oligomerize in the presence of its cognate Avr. However, given
that MLA13L11E/L15E forms stable heterocomplexes with AvrA13-1, we would
expect to observe a size increase of MLA13 due to effector binding. AvrA13-1-GFP is
approximately 40 kDa (as shown in the right panel), which should result in a
noticeable size shift of MLA13 in BN-PAGE upon binding to it. However, this
expected size shift is not evident (left panel, last three lanes). These data thus
support the authors' conclusion in one way but conflict in another. The authors
should provide a detailed explanation for:

a) Why is the expected size shift not observed upon AvrA13-1-GFP binding? Please
provide the immunoblot results using an anti-GFP antibody, showing where the
AvrA13-1-GFP signal is in BN-PAGE.

We repeated the experiment shown in Extended Data Fig. 6 from the original
manuscript, incorporating additional treatments and independent BN-PAGE analysis
as requested (please see Rebuttal Figures 1 and 2 below). We conducted sequential
immunoblotting with an anti-HA antibody to detect putatively non-oligomerized and
oligomerized forms of the MLA13 receptor, followed by stripping and reprobing of the
blot using anti-GFP antibody to detect the effectors. An AVR_{A13}-1-GFP effector signal
is detected, which corresponds in size to the mobility of the non-oligomerized MLA13
receptor (Rebuttal Figure 1B, arrowhead in lane 4), suggesting effector binding to the
non-oligomerized MLA13 receptor. By contrast, several AvrSr35-GFP effector signals
are detectable only corresponding to the mobility of multiple Sr35 oligomers
(Rebuttal Figure 1B, arrowheads in lane 1), indicating effector-bound Sr35 receptor
oligomers.

In the original Extended Data Fig. 6, the mobility of the non-oligomerized
MLA13-AVR_{A13} complex overlaps with the mobility of the receptor in the lane
containing MLA13 alone (original Extended Data Fig. 6 lane 5). The mobility of both
Sr35 and MLA13 resting state receptors in BN-BAGE is larger than the expected
molecular weight of the corresponding receptor monomers (each ~120 kDa as
shown by SDS-PAGE in Rebuttal Figure 2A and 2B, lower panels). The reason for
this remains unclear, but makes the separation of unbound and effector-bound non-
oligomerized receptors by BN-PAGE difficult.

As requested, we also tried to separate *N. benthamiana* leaf protein extracts
by BN-PAGE without immunoprecipitation. In these experiments we failed to detect
receptor and effector signals. We note that in the independent BN-PAGE analysis, an
additional Sr35 oligomer is detected at ~900 kDa (see rebuttal Figure 2B below),
which is close to the apparent molecular weight of the Sr35 resistosome (Förderer *et*
*al.*, 2022). This indicates that analysis by BN-PAGE results in variable Sr35
oligomers in independent experiments, presumably reflecting disassembly of the
Sr35 resistosome when using this method, compared to the stable pentameric
conformation observed when the proteins are passed through a soluble, mobile
phase by SEC (reformatted Appendix Figures S3 and S4).

Owing to variation of multiple Sr35 oligomers detected by BN-PAGE in
independent experiments and consistent undetectable MLA13 oligomerization when
co-expressed with AVR_{A13}-1, we propose to remove Extended Data Fig 6 from the
manuscript. Instead, we would like to highlight the SEC profiles presented in
Extended Data Fig. 7 from the original manuscript, which offer consistent data with
no evidence of heterocomplex disassociation. Extended Data Figure 7 has been
reformatted and is now included as Figure EV1.

Rebuttal Figure 1. Blue Native-PAGE assay testing the oligomeric states of MLA13 complexes.

(A) Purified protein samples were analysed by BN-PAGE (left top panel) with subsequent western blotting with an anti-HA antibody to detect receptors (MLA13 and Sr35). SDS-PAGE analysis of the input samples (bottom panel) was conducted to test the expression of input proteins. (B) The membrane was stripped and re-probed by anti-GFP antibody to detect the effectors (AVR_{A13}-1 and AvrSr35). The position of AVR_{A13}-1 and AvrSr35 is indicated by arrowheads.

Rebuttal Figure 2. Blue Native-PAGE assay testing the oligomeric states of MLA13 complexes.

(A) The result from Extended Data 7 of the original version of the manuscript. (B) Independent experiment was performed with the same design shown in (A).

b) How this result reconciles with the claim of stable heterocomplex formation.

As shown in new Rebuttal Figure 1 (see above), an AVR_{A13}-1-GFP effector signal is
detected, which corresponds in size to the mobility of the non-oligomerized MLA13
receptor (arrowhead in lane 4), suggesting effector binding to the non-oligomerized
MLA13 receptor. This provides independent evidence for the formation of a stable
heterocomplex by BN-PAGE and SEC analysis.

c) Or whether there are any technical limitations in the assay that might explain this
unexpected result.

As shown in new Rebuttal Figure 1, an AVR_{A13}-1-GFP effector signal is detected,
which corresponds in size to the mobility of the non-oligomerized MLA13 receptor
(arrowhead in lane 4), suggesting effector binding to the non-oligomerized MLA13
receptor. This provides independent evidence for effector binding to the non-
oligomerized MLA13 receptor by BN-PAGE and SEC analysis.

We cannot exclude that a hypothetical MLA13 resistosome disassembles into
receptor-effector heterodimers upon cell lysis and protein purification or is
inaccessible to our purification method. Regardless of whether the MLA13–AVR_{A13}-1
heterodimer results from disassembly of a hypothetical MLA13 resistosome or is a
ligand-bound intermediate state receptor, the established receptor-effector interface
is necessary for MLA13-mediated cell death. We have added this to the second
paragraph of the discussion.

3. Lines 222-224: The authors' interpretation of the MLA13/AvrA13-1 heterodimer
complex as an "intermediate state after effector binding-induced release of ADP but
before ATP binding-induced oligomerisation" requires further validation.

Our rationale for the MLA13^{K98E.K100E}–AVR_{A13}-1 heterodimer as a possible
intermediate state is based on the finding that the receptor conformation is
reminiscent of the nucleotide-free, ligand-bound intermediate complex of ZAR1–
RKS1 bound to PBL2^{UMP} (PDB: 6J5V). We wish to point out our use of cautious
conjecture, using 'may' or 'might.' As stated above, we cannot exclude that a
hypothetical MLA13 resistosome disassembles into receptor-effector heterodimers
upon cell lysis and protein purification or is inaccessible to our purification method.
Regardless of whether the MLA13–AVR_{A13}-1 heterodimer results from disassembly
of a hypothetical MLA13 resistosome or is a ligand-bound intermediate state
receptor, the established receptor-effector interface is necessary for MLA13-
mediated cell death. We have added this to the second paragraph of the discussion.

Unlike Zar1, MLA13/AvrA13-1 complex was expressed in the *N. benthamiana*
system that should provide all necessary components (including ATP) for full NLR
activation. It's unexpected that a functional NLR is stuck at an intermediate state but
not the final conformation. The authors should consider alternative explanations,
such as the possibility that the observed heterodimers are fragments of higher-order
oligomers that broke down during purification.

To conclusively demonstrate the *in vivo* state of the MLA13/AvrA13-1 complex and
avoid potential artifacts from tags and purification procedures, the authors should
conduct additional experiments:

a) Co-expressing functional MLA13/AvrA13-1 combinations (capable of triggering
cell death) in *N. benthamiana* and analyze these samples using BN-PAGE without
immunoprecipitation. This would allow detecting MLA13/AvrA13-1 in their native
state. Co-infiltration with LaCl₃ could be considered to suppress cell death to
enhance protein accumulation.

Thank you for the suggestion. In the revised manuscript we have added to the
discussion that we cannot exclude that a hypothetical MLA13 resistosome
disassembles into receptor-effector heterodimers upon cell lysis and protein
purification or is inaccessible to our purification method. It is also possible that the
concentration of a hypothetical MLA13 resistosome is too low compared to
heterodimers to be detected by our purification protocol, but still sufficient for
receptor-mediated cell death triggered by the effector.

As requested, we tried to separate *N. benthamiana* leaf protein extracts by
BN-PAGE without immunoprecipitation. In these experiments we failed to detect
receptor and effector signals. However, using BN-PAGE of immunoprecipitated
protein, we have verified in Rebuttal Figure 1 effector binding to the non-
oligomerized MLA13 receptor.

b) Checking whether adding ATP or dATP to the purified MLA13/Avr13A-1
heterodimers triggers oligomerization.

As the reviewer has pointed out above, the use of *N. benthamiana* as an expression
system provides all the necessary molecules, including ATP, for cell death activity
and putative receptor oligomerisation. That said, we previously attempted adding
ATP to the purified heterodimer to enforce oligomerisation of a high-order complex,
however, the samples persistently eluted from SEC at the same elution volume as
the structurally-resolved heterodimer. Importantly, the successful purification of both
Sr35 and Sr50 resistosomes (Extended Data Figs. 4 and 5 of the original version of
the manuscript; Appendix Figures S3 and S4 in the revised version of the
manuscript) using the same protein extraction and purification method verifies that all
necessary components are present *in planta* for the formation and purification of a
hypothetical MLA13 resistosome and that our method can be used to purify at least
these two sensor CNL resistosomes as pentameric assemblies.

c) Extracting the nucleotide binding to the heterodimers and check its identity by
HPLC.

We have thoroughly checked the MLA13-AVR_{A13-1} Coulomb potential map, and it
lacks density in the canonical nucleotide binding site. This finding is mentioned in
line 219 of the original manuscript and is shown in the provided Coulomb potential
map.

4. Fig.3 and 4 Please include the corresponding interaction analyses, such as co-IP,
split-luciferase assay, or BiFC, for all the residue substitution analyses shown in
Figures 3 and 4. These additional tests are essential to verify that the observed loss
of cell death resulting from residue substitutions is indeed due to a disruption in
interaction, rather than other potential factors, such as inhibiting conformational
changes.

We have conducted co-IP experiments to address the point raised by the reviewer.
 The results of co-IPs of select AVR_{A13}-1 substitution mutants that result in a loss of
 cell death phenotype are shown in new Appendix Figure S6. Immunoprecipitation *via*
 the Twin-Strep-tag on wild-type AVR_{A13}-1 or AVR_{A22} or four tested high-order AVR_{A13}-
 1 mutants, robustly coprecipitated with MLA13-4×MYC receptor only when co-
 expressed with wild-type AVR_{A13}-1 (Appendix Figure S6).

Appendix Figure S6. Co-IP assays of AVR_{A13}-1 interface substitution mutants that result in a reduced or loss of cell death activity.

Protein was immunoprecipitated *via* the Twin-Strep-tag on AVR_{A13}-1. Samples were run on a 12% SDS PAGE gel.

5. The engineering part: The authors describe how they identified the residues MLA7(L902) and MLA13(S902) as potentially responsible for the differential recognition between MLA7 and MLA13.

They note that the residue-switched variant MLA7L902S can confer new recognition of Avr13A-1. But the mechanism such as how it works is not well explained.

We appreciate this comment. We explain how this substitution may gain recognition of AVR_{A13}-1 and AVR_{A13}-V2 in lines 382-392 of the original version of the manuscript. However, due to the molecular complexity of the putative MLA7^{L902S}-AVR_{A13}-1 interface, a high-resolution structure of the MLA7^{L902S}-AVR_{A13}-1 interface would be necessary to make experimental, evidence-based statements about this interaction.

When examining Fig. 4, I noticed that MLA13(S902) actually contributes little to its recognition of Avr13A-1. Specifically, the mutant variant MLA13S902A still triggers strong cell death when co-expressed with Avr13A-1 (Figure 4C).

We only conclude from these data that co-expression of the single substitution mutant MLA13^{S902A} with AVR_{A13}-1 does not lead to a loss of cell death. This does not

exclude the possibility that MLA13^{S902} does contribute to the broader receptor-
effector interface. In other words, if MLA13^{S902} was combined with other MLA13
substitutions that result in a loss of cell death (i.e., MLA13^{R938A}), cell death activity
may be further impeded.

This observation raises questions about the mechanism by which the MLA7L902S
swap functions:

Thank you for the comment. We do not claim and the data does not suggest that
MLA7^{L902S} swaps functions, but rather gains cell death activity when co-expressed
with AVR_{A13-1} and AVR_{A13-V2} while retaining cell death activity when co-expressed
with AVR_{A7} variants.

a) Does Leu902 in MLA7 cause steric clashes with Avr13A-1, thereby preventing
interaction? And the substitution to a smaller residue like Serine remove this clash
and allow recognition?

b) To test this hypothesis, the authors could mutate MLA7(L902) to other small
residues with short side chains and assess whether these variants can recognize
Avr13A-1. Additionally, they could explore the effects of substituting MLA13(S902)
with Leu or other large side chain residues, or with other small side chain residues.

To address this suggestion by the reviewer, we generated and tested additional
receptor variants. The MLA13^{S902L} variant was co-expressed with AVR_{A13-1} in barley
protoplasts and leaves of *N. benthamiana*. These data have been added to Figure 4
of the revised version of the manuscript. We also generated MLA7^{L902A} and co-
expressed it with AVR_{A22}, AVR_{A7-1}, AVR_{A7-2}, AVR_{A13-1} and AVR_{A13-V2} (Appendix
Figure S8 in the revised version of the manuscript). We have added a sentence in
the text of the revised manuscript describing that the MLA7^{L902A} variant phenocopies
MLA7^{L902S}.

c) Finally, it is essential to test the interactions of all these variants, either by co-IP,
split-luciferase assay, or by BiFC.

We performed co-IP experiments with selected AVR_{A13-1} substitution mutants that
result in a loss of cell death phenotype with MLA13 as shown above and in new
Appendix Figure S6.

During the revision of our manuscript, a preprint from another laboratory has
been posted that reports the results of *in vitro* shuffling of barley *Mla* genes
(<https://www.biorxiv.org/content/10.1101/2024.10.27.619561v1>). These authors
show that after recombination of *Mla7* and *Mla13* genes *in vitro* and subsequent
screening of the resulting library, substitution of the exact same amino acid
substitution in MLA7 that we had engineered based on our structural knowledge
facilitated interaction with AVR_{A13-1} in yeast and AVR_{A13-1}-dependent immune
signalling *in planta*. This independent study reports yeast interaction data for the
MLA7^{L902S} gain-of-recognition receptor variant with the virulent version of AVR_{A13-V2}
in Fig. 4, which extends and supports the results of our cell death assays *in planta*
using the equivalent receptor and effector variants shown in Fig. 5 of our manuscript.

We emphasise that, consistent with current models implicating steric clash
with the NBD as conserved activation mechanism for sensor CNLs (Förderer *et al.*,
*Current Opinion in Plant Biology*, 2022), binding to MLA receptors can be insufficient

for receptor activation. For example, some *Bg* AVR_{As} are able to bind MLAs without
triggering a cell death response (Saur *et al.*, *eLIFE* 2019; Crean *et al.*, *J of*
*Experimental Botany* 2023).

Other comments:

1. Extended Data Fig. 3 Despite the significant difference in size between myc and
GST tags, myc-MLA13 variants in Extended Data Fig. 3b appear to have similar
molecular weights to GST-MLA13 shown in Extended Data Fig. 1. Could the author
explain this and give a bit more detail of how they performed these experiments?
Improved descriptions in the legends are necessary for better clarity and
understanding.

Western blots and CBB-stained gels and western blots throughout the manuscript
were run on gels of different percentages. The same molecule can migrate slightly
differently on gels with differing percentages of polyacrylamide
([https://www.sigmaaldrich.com/DE/de/technical-documents/technical-article/protein-](https://www.sigmaaldrich.com/DE/de/technical-documents/technical-article/protein-biology/western-blotting/observed-vs-calculated-molecular-weight?srsId=AfmBOoo1J3Ku90qEtm_W2m11XEn4xJenqSNSrKYB-WdkxLVzJaf0lo7j)
[biology/western-blotting/observed-vs-calculated-molecular-](https://www.sigmaaldrich.com/DE/de/technical-documents/technical-article/protein-biology/western-blotting/observed-vs-calculated-molecular-weight?srsId=AfmBOoo1J3Ku90qEtm_W2m11XEn4xJenqSNSrKYB-WdkxLVzJaf0lo7j)
[weight?srsId=AfmBOoo1J3Ku90qEtm_W2m11XEn4xJenqSNSrKYB-](https://www.sigmaaldrich.com/DE/de/technical-documents/technical-article/protein-biology/western-blotting/observed-vs-calculated-molecular-weight?srsId=AfmBOoo1J3Ku90qEtm_W2m11XEn4xJenqSNSrKYB-WdkxLVzJaf0lo7j)
[WdkxLVzJaf0lo7j](https://www.sigmaaldrich.com/DE/de/technical-documents/technical-article/protein-biology/western-blotting/observed-vs-calculated-molecular-weight?srsId=AfmBOoo1J3Ku90qEtm_W2m11XEn4xJenqSNSrKYB-WdkxLVzJaf0lo7j)). The methods used for protein purification and western blotting
are described in the methods section. We have now indicated gel percentages
where possible in the revised manuscript.

2. Fig1a The figure is confusing and requires improvement in labelling and clarity.
The authors state that the inserted SDS-PAGE gel represents "fractions eluted along
the black line". However, the placement of the 875 kDa arrow at the end of the black
line suggests that the samples are loaded in reverse order - from large elution
volume to small. This contradiction needs clarification.

Thank you for your recommendations. We reformatted Figure 1A and changes to the
figure legend are highlighted in yellow.

3. Fig. 2C, D Please provide the prediction scores (pLDDT and PAE plots, pTM and
ipTM) for AF3 models.

The requested data is provided in Appendix Figure S5.

4. Fig. 3e Data here is expected to provide evidence that MLA13 was expressed
among all the panels in Fig. 3C, especially the combinations losing cell death. The
current figure is meaningless. Same issue with Fig. 4e.

Figures 3E and 4E in the original version of the manuscript have now been omitted.

5. Fig3a and 4a The separation of interacting residues between the receptor and
effector in Fig. 3a (effector side) and Fig. 4a (NLR side) makes it difficult to
comprehend the complete interaction interface. While this approach keeps each
figure tidy, it hinders understanding of which NLR residues interact with specific
effector residues. An additional merged figure is recommended, showing both NLR
and effector residues involved in the interaction.

We edited Figure 3A to only highlight residues with atom display that we
experimentally tested in cell death assays. We added a domain architecture
schematic below Figure 4A to make the figure more accessible to the readership. In
addition, we included a figure (Figure EV3 in the revised version of the manuscript)
to show all the experimentally tested interface residues in two views. We hope that
this fulfils the reviewer's recommendation.

In addition:

• What is the net molecular weight of the heterodimer used in this structural study in
499 kDa? This has to be included somewhere in the paper as this is a cryoEM study of
500 intermediate size protein complex.

The calculated molecular weight of the GST-MLA13^{K98E/K100E}-AVR_{A13}-1 heterodimer
is ~155 kDa as noted in Figure 1A of the original version of the manuscript. We have
added this to the Figure 1 legend for further clarification.

• Where does K98 and K100 come in the 3D structure of monomeric ZAR1? It is in
the CC domain, but the exact location of these residues and their interactions must
be discussed. For example, are these residues lying in the allosteric path that leads
to fold switch of CC domain N-terminal alpha1 helix? Are they interacting with the
alpha1 helix in the monomeric ZAR1? These details will clarify the questions on
whether the mutation K98E, K100E impacts the conformational transition required to
release the alpha1 helix after effector recognition and assemble into resistosome in
an autoactive MLA D502V mutant.

The CC domains of MLA13 and ZAR1 are highly polymorphic, making it
unreasonable to draw concrete conclusions from sequence comparisons. Moreover,
the aforementioned stretch of sequence is unresolved in all currently available
resistosome structures, again making it difficult to draw conclusions on the role of
MLA13^{K98/K100} in a hypothetical resistosome. Nevertheless, although co-expression of
MLA13^{K98E/K100E} with AVR_{A13}-1 results in a loss of cell death, the addition of the
autoactive substitution (MLA13^{K98E/K100E/D502V}) results in a rescue of cell death activity
(Extended Data Fig. 3A of the original version of the manuscript), suggesting that the
substitutions MLA13^{K98E/K100E} do not interfere with the formation of a cell death-
inducing conformation (i.e. hypothetical resistosome formation). This is discussed in
lines 130-135 of the original version of the manuscript. Please see Figure EV1 of the
revised version of the manuscript. Discussing the physiological roles of residues
MLA13^{K98/K100} with the currently available literature and evidence would be
speculative.

• In Line 137, the authors suggest that analogous substitution in NRG1 retains
oligomerisation. Although ZAR1, MLA13 and NRG1 are CNLs, we know that NRG1
forms a phylogenetically different class of RPW8-NLRs, so their resistosome
assembly might be different from that of ZAR1 or MLA13. It is possible that they form
similar resisting state structure but have a different resistosome structure.

We appreciate the comment and agree with the reviewer that a structure of an
activated RPW8-NLR is needed to clarify whether its oligomeric state and structure
is similar or dissimilar to ZAR1 and MLA13.

• In Line 146-147, the authors suggest that SEC revealed a heterocomplex with size
smaller than multimeric MLA13. Since there is cell death in this triple mutant, it is
possible that the multimers are just at very low concentration.

We did not purify and resolve the triple mutant, autoactive MLA13
(MLA13^{K98E/K100E/D502V}). We resolved and report the structure of MLA13^{K98E/K100E}_
AVR_{A13}-1. This is discussed in line 135 of the original version of the manuscript.

Would it be a good idea in that case, to pull-down the MLA13 effector complex first
with MLA's flag-tag and screen in negative stain before going for second pull-down
from effector side? Such negative stain images immediately after first pull-down from
crude extract without SEC may reveal a lower number of existing reistosomes,
though not homogeneous.

Thank you for the comment. We did not use a flag tag in any of the reported
experiments. As requested, we purified a large-scale sample of autoactive
MLA13^{L11E/L15E/D502V} and analysed the sample by negative staining TEM after a single
step affinity purification only. We did not detect the presence of a high-order complex
or star-shaped particles in this sample (Figure EV1 (SEC profiles) and Rebuttal
Figure 3 (TEM image)).

**Rebuttal Figure 3. Representative negative staining TEM image of the purified**
**MLA13^{L11E/L15E/D502V}-2S-HA.**

The sample was analysed by negative staining TEM directly after the first-step affinity purification
before SEC analysis. Purification result shown in Expanded View Figure 1A. Black scale bar at the
bottom represents 100 nm.

In Figure 2, the legend describes as 'electron density map'. The authors know very
well that the map from cryo-EM reconstruction is a 'coulomb potential map' and not
electron density map as in X-ray crystallography. This typing mistake has to be
corrected.

Thank you for your correction. The terminology has been changed throughout the
revised version of the manuscript.

• In Line 271 and 275, MLA13 residue 934 is given as F and Y respectively. The
atomic model displayed in the Figure 4A, shows it as Y934. This typing mistake has
to be corrected in Line 271.

Thank you for your correction. The mistake has been corrected in the revised version
of the manuscript.

• Here is a general observation in extended figure 6, left panel. The last but one lane,
where MLA13 alone is loaded, runs in the BN-PAGE near 242kDa, but its molecular
weight is around 107kDa for single chain. Such migration of the apoprotein in resting
state is an indication for a dimeric state in resting state as observed for NRC2.
Resting NRC2 also migrates near this location in BN-PAGE since they are dimers.
The authors can track down this if they are interested in resolving the resting state of
MLA.

Thank you for the comment. In the original Extended Data Fig. 6, the mobility of the
non-oligomerized MLA13-AVRa13 complex overlaps with the mobility of the receptor
in the lane containing MLA13 alone (original Extended data Fig. 6 lane 5). The
mobility of both Sr35 and MLA13 resting state receptors in BN-BAGE is larger than
the expected molecular weight of the corresponding receptor monomers (each
approx. 120 kDa as shown above by SDS-PAGE in Rebuttal Figure 2A and 2B; see
also above). The reason for this remains unclear, but makes the separation of
unbound and effector-bound non-oligomerized receptors by BN-PAGE difficult.

• In the Table 1 of extended data, there is no description for 'Energy filter' is given nor
mentioned in the cryo-EM imaging section. Small protein complexes normally benefit
from the use of energy filter during imaging. Was there any particular reason why it
was not used in this imaging experiment? If there is any advantage, information
along this will be useful for all others working on this range of molecular size.

We respectfully disagree with the statement that small proteins usually benefit from
the use of an energy filter during imaging. We agree that there is a correlation
between deposited high-resolution structures of small proteins and the use of energy
filters, but believe this to be mainly due to the fact that the first widely used direct
electron detector was often sold with an energy filter. In fact, since the advantage of
energy filters is to remove inelastically scattered electrons, they are more useful for
larger complexes, where the layer of vitreous ice must be thick to fully support the
sample, or for tomography, where the sample must be imaged at higher tilt settings
that increase the cross section of vitreous ice that the beam must pass through. In
the case of this manuscript, an energy filter or its settings were not mentioned
because the microscope used for data acquisition is not equipped with an energy
filter. However, we have shown (Vu H.H., Behrmann H., *et al.* 2023) that this
microscope is perfectly suitable for imaging a much smaller protein complex of 120
634 kDa at a resolution of 2.6 Å.

• Finally, the heterocomplex structure describes how this NLR is engaging its
different parts to recognize its cognate effector. Does the authors think, the effector
position and the recognition areas of NLR change when this heterocomplex at non-
canonical conformation moves to oligomeric, resistosome conformation. Can the

initial recognition mode of the effector and its final position in assembled resistosome
be expected to vary? All views of the authors along this will be useful in the
discussion on engineering perspectives.

Please note that we lack experimental evidence for a hypothetical, effector-triggered
MLA13 resistosome. To avoid misinterpretation by the readership, we wish to refrain
from discussing whether and how the effector position could change when the
heterodimeric complex transitions to a hypothetical resistosome.

Referee #2:

In this study, the authors show the structural basis of MLA13 recognition of AVR_{A13}
and manage to modify the specificity of the NLR which is very impressive. The
description of NLR/effector recognition is well supported and very convincing.

I have one major point regarding the biological relevance of the MLA13 K98/K100
mutant used in this study.

Is this MLA13 heterodimer active and does it reflect the function of the WT protein?

Thank you for the comment. The purification experiments of several MLA-AVR_A
variants (Figure EV1 of the revised manuscript), including untagged MLA13^{L11E/L15E},
show that all elute at the same volume as the structurally resolved heterodimer or
later in the profile without evidence for the formation of a high-order MLA13-AVR_{A13-1}
1 complex. We generated interface substitution mutants based on the heterodimer
structure that resulted in a loss of cell death response upon co-expression in both
barley protoplasts and leaves of *N. benthamiana*. These results suggest that the
interface is biologically relevant. Our findings are further supported by an
independent study from another laboratory that reports the results of *in vitro* shuffling
of barley *Mla* genes (<https://www.biorxiv.org/content/10.1101/2024.10.27.619561v1> ;
posted during the revision of our manuscript). Remarkably, these authors show that
after recombination of wild-type *Mla7* and *Mla13* genes *in vitro* and subsequent
screening of the resulting library, substitution of the exact same amino acid
substitution in wild-type MLA7 that we had engineered based on our structural
knowledge facilitated interaction with AVR_{A13-1} in yeast and AVR_{A13-1}-dependent and
receptor-mediated cell death *in planta*. Together, this strongly supports the biological
relevance of our structure-based study for understanding effector-triggered MLA-
mediated cell death activation.

Does the WT protein exist *in vivo* as a heterodimer similar to the one seen here?

Do we see higher order oligomers of MLA13 WT in *planta* following AVR13
recognition?

We refer to our response to the preceding point raised by the reviewer.

Unfortunately, the oligomeric state of the wild-type MLA13-AVR_{A13-1} complex *in vivo*
remains undefined as co-expression of receptor and effector results in a rapid cell
death response and depletion of detectable protein. For this reason, we present
results of cell death-impeding substitutions in MLA13 that allow abundant
resistosome formation of both Sr35 and Sr50 CNLs in *N. benthamiana* using the
exact same protocol that resulted in purification of the MLA13-AVR_{A13-1} heterodimer
only (Appendix Figures S3 and S4). In Extended Data Fig. 7 of the original version of

the manuscript we present purification results of a number of different MLA13-
AVR_{A13}-1 substitution and tag combinations along with the positive controls Sr35 and
Sr50 resistosomes. Please see Figure EV1 of the revised version of the manuscript
for a reformatted version of Extended Data Fig. 7 from the original version of the
manuscript.

The claim that this dimeric state represents an intermediate step of MLA13 activation
is weakly supported by data.

Our rationale for the MLA13^{K98E.K100E}-AVR_{A13}-1 heterodimer as a possible
intermediate state is based on the finding that the receptor conformation is
reminiscent of the nucleotide-free, ligand-bound intermediate complex of ZAR1-
RKS1 bound to PBL2^{UMP} (PDB: 6J5V). Due to the unexpected heterodimeric
conformation of the MLA13-AVR_{A13}-1 heterocomplex, we have limited our
conclusions in the original manuscript to cautious conjecture. We do not make claims
about the physiological function of the heterodimeric conformation. Our conclusions
regarding the MLA13^{K98E/K100E}-AVR_{A13}-1 heterodimer and its biological relevance are
limited to structure-guided substitutions at the effector-binding interface that result in
a loss of cell death. There are three possible explanations for the finding of a stable
MLA13^{K98E/K100E}-AVR_{A13}-1 heterodimeric conformation, which we have now included
in the discussion of the revised manuscript: "First, a hypothetical MLA13 resistosome
may disassemble into receptor-effector heterodimers upon cell lysis and protein
purification or is otherwise inaccessible to our purification method. Second, the
concentration of a hypothetical MLA13 resistosome is too low compared to the
heterodimers to be isolated by our purification protocol, but still sufficient for
receptor-mediated cell death triggered by the effector. Third, the stable
MLA13^{K98E/K100E}-AVR_{A13}-1 heterodimer represents a non-pentameric, novel
immunostimulatory output. Of note, regardless of whether the MLA13^{K98E/K100E}-
AVR_{A13}-1 heterodimer results from disassembly of a hypothetical MLA13
resistosome or is a ligand-bound intermediate state receptor with immunostimulatory
activity, the observed receptor-effector interface is necessary for MLA13-mediated
cell death."

The authors should provide information on the function of MLA13 K98E/K100E
compared to WT. Ideally, the authors should show that the WT protein form higher
order oligomers following AVRA13 recognition, that the K98/K100 D502V forms
similar oligomers and determine the structure of the K98/K100 D502V mutant
oligomer.

Although co-expression of MLA13^{K98E/K100E} with AVR_{A13}-1 results in a loss of cell
death, the addition of the autoactive substitution (MLA13^{K98E/K100E/D502V}) results in a
rescue of cell death (Extended Data Fig. 3A of the original version of the
manuscript), suggesting that the substitutions MLA13^{K98E/K100E} do not block a cell
death-inducing conformation (i.e. hypothetical resistosome formation). This is
discussed in lines 130-135 of the original version of the manuscript. We have no
evidence for the formation of high-order MLA13-AVR_{A13}-1 oligomers, and therefore
cannot show this. In Extended Data Fig. 7 of the original version of the manuscript
we present purification results of a number of different MLA13-AVR_{A13}-1 substitution
and tag combinations along with the positive controls Sr35 and Sr50 CNL
resistosomes that were purified using the exact same protocol (Appendix Figures S3

and S4). Please see Figure EV1 of the revised version of the manuscript for a
reformatted version of Extended Data Fig. 7 from the original version of the
manuscript.

Is MLA13 WT also sensitive to calcium channel inhibitors? At least, the author should
provide data on MLA13 K98/100 D502V oligomerization status.

Please justify the use of the L11/15 D502V mutant for the determination of
oligomerization in blue native gels.

Thank you for the comment. Effector-triggered MLA13-mediated cell death is indeed
sensitive to LaCl_3 inhibition as shown in an earlier publication (Fig. 5A, Crean *et al.*,
2023). We have used MLA13^{L11E/L15E} for BN-PAGE to determine the oligomerization
status of MLA13^{L11E/L15E/D502V} because the corresponding substitutions in Sr35^{L11E/L15E}
and Sr50^{L11E/L15E} have been shown to retain the formation of resistosomes (Förderer
*et al.*, 2022; unpublished) and MLA13^{L11E/L15E} forms a stable heterodimer with
AVR_{A13}-1 that is indistinguishable from the MLA13^{K98E/K100E} heterodimer with the
effector (Figure EV1).

We repeated the experiment shown in Extended Data Fig. 6 from the original
manuscript, incorporating additional treatments and independent BN-PAGE analysis
as requested (Rebuttal Figures 1 and 2, shown below). We conducted sequential
immunoblotting with an anti-HA antibody to detect putatively non-oligomerized and
oligomerized forms of the MLA13 receptor, followed by stripping and reprobing of the
blot using anti-GFP antibody to detect the effectors. An AVR_{A13}-1-GFP effector signal
is detected, which corresponds in size to the mobility of the non-oligomerized MLA13
receptor (Rebuttal Figure 1B, arrowhead in lane 4), suggesting effector binding to the
non-oligomerized MLA13 receptor. By contrast, several AvrSr35-GFP effector signals
are detectable only corresponding to the mobility of multiple Sr35 oligomers
(Rebuttal Figure 1B, arrowheads in lane 1), indicating effector-bound Sr35 receptor
oligomers.

In the original Extended Data Fig. 6, the mobility of the non-oligomerized
MLA13-AVR_{A13} complex overlaps with the mobility of the receptor in the lane
containing MLA13 alone (original Extended Data Fig. 6 lane 5). The mobility of both
Sr35 and MLA13 resting state receptors in BN-BAGE is larger than the expected
molecular weight of the corresponding receptor monomers (each approx. 120 kDa as
shown by SDS-PAGE in Rebuttal Figure 2, lower panels). The reason for this
remains unclear, but makes the separation of unbound and effector-bound non-
oligomerized receptors by BN-PAGE difficult.

As requested, we also tried to separate *N. benthamiana* leaf protein extracts
by BN-PAGE without immunoprecipitation. In these experiments we failed to detect
receptor and effector signals. We note that in the independent BN-PAGE analysis, an
additional Sr35 oligomer is detected at ~900 kDa (Rebuttal Figure 2B shown below),
which is close to the apparent molecular weight of the Sr35 resistosome (Förderer A.
*et al.*, 2022). This indicates that analysis by BN-PAGE results in variable Sr35
oligomers in independent experiments, presumably reflecting disassembly of the
Sr35 resistosome when using this method, compared to the homogenous
conformation observed when the molecule is passed through a soluble, mobile
phase by SEC (Figure EV1).

Owing to variation of multiple Sr35 oligomers detected by BN-PAGE in
independent experiments and consistent undetectable MLA13 oligomerization when
co-expressed with AVR_{A13}-1, we propose to remove Extended Data Fig 6 from the

manuscript. Instead, we would like to highlight the SEC profiles presented in
 Extended Data Fig. 7 from the original manuscript, which offer consistent data with
 no evidence of heterocomplex disassociation. Extended Data Figure 7 has been
 reformatted and is now included as Figure EV1.

**Rebuttal Figure 1. Blue Native-PAGE assay testing the oligomeric states of MLA13**
 **complexes.**

**(A)** Purified protein samples were analysed by BN-PAGE (left top panel) with subsequent western
 blotting with an anti-HA antibody to detect receptors (MLA13 and Sr35). SDS-PAGE analysis of
 the input samples (bottom panel) was conducted to test the expression of input proteins. **(B)** The
 membrane was stripped and re-probed with anti-GFP antibody to detect the effectors (AVR_{A13}-1 and
 AvrSr35). The position of AVR_{A13}-1 and AvrSr35 is indicated by arrowheads.

**Rebuttal Figure 2. Blue Native-PAGE assay testing the oligomeric states of MLA13**
 **complexes.**

**(A)** The result from Extended Data 7 of the original version of the manuscript. **(B)** Independent
 experiment was performed with the same design shown in **(A)**.

Why did the authors use the K98/100 mutant for the structural study instead of the
L11/15 mutant?

We used MLA13^{K98E/K100E} because MLA13^{L11E/L15E} results in the accumulation of
notably lower protein levels. In Extended Data Fig. 7 of the original version of the
manuscript we present purification results of a number of different MLA13-AVR_{A13-1}
substitution and tag combinations along with the positive controls (Sr35 and Sr50
resistosomes; Appendix Figures S3 and S4) that were purified using the same
method. Co-expression of MLA13^{L11E/L15E}- AVR_{A13-1} resulted in an elution volume
comparable to MLA13^{K98E/K100E}- AVR_{A13-1}. Extended Data Figure 7 has been
reformatted and is now included as Figure EV1.

This mutant makes higher order oligomers similar to active resistosomes.

In Extended Data Fig. 7 of the original version of the manuscript we present
purification results of a number of different MLA13-AVR_{A13-1} substitution and tag
combinations along with the positive controls (Sr35 and Sr50 resistosomes;
Appendix Figures S3 and S4) that were purified using the same method. Co-
expression of MLA13^{L11E/L15E}- AVR_{A13-1} resulted in an elution volume comparable to
MLA13^{K98E/K100E}- AVR_{A13-1}. Extended Data Figure 7 has been reformatted and is
now included as Figure EV1.

It is unclear in extended figure 6 if the D502V mutation is combined with the L11/15
mutations. Please specify in the caption.

As explained above, we used MLA13^{L11E/L15E} for BN-PAGE to determine the
oligomerization status of MLA13^{L11E/L15E/D502V} because the corresponding
substitutions in Sr35^{L11E/L15E} and Sr50^{L11E/L15E} have been shown to retain the
formation of resistosomes (Förderer *et al.*, 2022; unpublished data for Sr50). Co-
expression of MLA13^{L11E/L15E}- AVR_{A13-1} resulted in an elution volume comparable to
MLA13^{K98E/K100E}- AVR_{A13-1} (Extended Data Figure 7 of the original manuscript,
reformatted as Figure EV1 in the revised manuscript.

As explained above, we propose to remove Extended Data Fig 6 from the
manuscript. Instead, we would like to highlight the SEC profiles presented in
Extended Data Fig. 7 from the original manuscript, which offer consistent data with
no evidence of complex disassociation. Extended Data Figure 7 has been
reformatted and is now included as Figure EV1.

Please specify in the figures when MLA13 K98/100 mutants are used. This
information is essential.

Thank you for the suggestion. We have now provided this information throughout the
revised version of the manuscript.

Where are residues L98/100 used for purifying MLA13? How do they affect MLA13
function? Please provide a view of the structure with these residues highlighted. Also
in figure 1, please indicate the number of amino-acids comprised in the different
domains of MLA13 in the schematic below the structure.

Thank you for your recommendation. We have clarified this in Figure 1 of the revised
version of the manuscript.

Another major point. How does the purification process affect the complex retrieved?
Do you observe the same SEC profile by pulling down MLA13? Does AVRA13
interact with oligomeric forms of MLA13 such as the one seen in extended figure 6?

In Extended Data Fig. 7 of the original version of the manuscript we present
purification results of a number of different MLA13-AVR_{A13}-1 substitution and tag
combinations along with the positive controls Sr35 and Sr50 resistosomes that were
purified using the same method. This resulted in a consistent purification of the
MLA13-AVR_{A13}-1 heterodimer, even when pulling down MLA13 without a second IP
via the effector tag. In addition, the oligomeric form of MLA13 shown in extended
Figure 6 is an autoactive receptor expressed in the absence of AVR_{A13}-1. We never
detect MLA13 multimers using BN-PAGE or SEC analysis when receptor and
effector are co-expressed. Please see an updated version of this figure as Figure
EV1 in the revised version of the manuscript.

If not, maybe the purification process enriches heterodimers and not full
resistosomes?

We agree with this possibility and have included this in the discussion of the revised
version of the manuscript. However, we emphasise that both Sr35 and Sr50 CNL
resistosomes were successfully purified from *N. benthamiana* in our study using
exactly the same expression and protein purification protocol.

Minor point:

Line 133, figure numbering is off, extended figure 3 is cited before extended figure 1.

Thank you for your correction. We have changed the corresponding figures and
numbering of the figures.

Referee #3:

This manuscript presents the cryo-EM structure of barley CNL immune receptor
MLA13 in complex with its cognate effector AVRA13-1 as a heterodimer. In contrast
to previously reported effector activated CNL pentamers, they propose MLA13-
AVRA13-1 heterodimer as an intermediate state.

They validated the importance of the interactions between MLA13-AVRA13-1 for
function. They further engineered a single amino acid substitution in MLA7 that
enables expanded effector detection of AVRA13-1 and the virulent variant AVRA13-
V2. This study significantly advanced the understanding on barley MLA receptors
and their recognition of effectors. The manuscript is well written. I have a few
comments for the authors to improve the manuscript.

1, In extended data Fig. 6, BN-PAGE showed the putative oligomer of
MLA13L11E/L15E/D502V. Can the authors also purify MLA13L11E/L15E/D502V and
do negative stain of it as they did for Sr35 resistosome?

Our revision experiments include the purification and negative staining imaging of
MLA13^{L11E/L15E/D502V} to address the comment of this and the other reviewers (Figure
EV1; Rebuttal Figure 3). Purification and negative stain imaging of
MLA13^{L11E/L15E/D502V} does not result in the elution of a high-order molecule from SEC
or resistosome-shaped particles from negative staining analysis, suggesting that a
hypothetical MLA13^{L11E/L15E/D502V} resistosome is not purified using our method.

**Rebuttal Figure 3. Representative negative staining TEM image of the purified**
**MLA13^{L11E/L15E/D502V}-2S-HA.**

The sample was analysed by negative staining TEM directly after the first-step affinity purification
before SEC analysis. Purification result shown in Expanded View Figure 1A. Black scale bar at the
bottom represents 100 nm.

2, For Nb cell death images, can the authors please make a mock leaf showing the
infiltration settings for readers to better understand?

Thank you for your recommendation. We have added leaf 'maps' to these figures
and moved them to 'Source Data'.

3, In extended data Fig. 1, please run all samples on one gel.

Thank you for your recommendation. These gels have now been replaced with those
from the same experiment performed on a different day where the same percentage
of SDS-PAGE gel was used. We did not perform the experiment again as infiltrating,
processing and purifying 200 grams of leaf tissue to simply run the two samples on
the same gel would be an unreasonable use of resources. The revised figure is now
listed as Appendix Figure S2.

Dear Paul,

We have now received re-review reports from all three referees, which I have included below. As you will see, you have addressed their concerns satisfactorily. Before I can finally accept the manuscript, there are some remaining editorial points which need to be addressed. In this regard would you please:

- include the corresponding authors' email addresses on the manuscript title page,
- select five keywords,
- reformat the reference list to alphabetical order instead of numbered; in the reference section, for longer author lists use the format of 10 authors + et al.,
- rename the conflict of interests statement the "Disclosure and competing interests statement",
- remove the AC/CrediT section from the text,
- remove the manuscript callout for Supplementary Table 1 (no such table is uploaded), include callouts in the manuscript for Figure panels 3D, 4A, 4D, 5a, and figures EV1, EV4 and Appendix Figures,
- upload main and EV figures as individual, high-resolution figure files, with their legends included in the manuscript file below the reference section,
- convert the appendix file to PDF format, include a title page containing "Appendix for The barley MLA13-AVRA13 heterodimer reveals principles for immunoreceptor recognition of RNase-like" and a table of contents with page numbers of the included items,
- upload a Tools and Reagents table as an individual file using the template from our guide to authors,
- indicate separate stained membranes and blots (which are currently spliced together) in figure 3D, figure 4D with a black line,
- provide specific URLs for 9FYC, EMD-50863 datasets in the data availability statement,
- provide exact p values in the legends of figures 3B, 4B and 5A,
- indicate the statistical test used for data analysis in the legends of figures 3B and 4B,
- define box plots in terms of minima, maxima, centre, bounds of box and whiskers, and percentile in the legends of figures 3B, 4B, 5A, and
- correct the section order as follows: Title page - Abstract & Keywords - Introduction - Results - Discussion - Methods - Data Availability - Acknowledgements - Disclosure and Competing Interests Statement - References - Figure Legends - Table(s) - Expanded View Figure Legends.

We include a synopsis of the paper (see <http://emboj.embojpress.org/>). Please provide me with a general summary image, a two sentence statement and 3-5 bullet points that capture the key findings of the paper.

I am looking forward to receiving your revised manuscript.

EMBO Press is an editorially independent publishing platform for the development of EMBO scientific publications.

Best wishes,

William

William Teale, PhD
Editor
The EMBO Journal
w.teale@embojournal.org

See also figure legend guidelines: <https://www.embojpress.org/page/journal/14602075/authorguide#figureformat>

- a point-by-point response to the referees' comments, with a detailed description of the changes made (as a word file).
- a word file of the manuscript text.
- individual production quality figure files (one file per figure)

- a complete author checklist, which you can download from our author guidelines (<https://www.embopress.org/page/journal/14602075/authorguide>).

- Expanded View files (replacing Supplementary Information)

We realize that it is difficult to revise to a specific deadline. In the interest of protecting the conceptual advance provided by the work, we recommend a revision within 3 months (14th Apr 2025). Please discuss the revision progress ahead of this time with the editor if you require more time to complete the revisions. Use the link below to submit your revision:

Referee #1:

The authors have satisfactorily addressed the issues we raised in the previous review

Referee #2:

The authors answered all my comments and I am satisfied with this version of the manuscript.
Thank you for your efforts.

Referee #3:

I have no more comments or concerns.

All editorial and formatting issues were resolved by the authors.

Dear Paul,

I am pleased to inform you that your manuscript has been accepted for publication in the EMBO Journal.

Congratulations to you and your team on a really insightful study!

Yours sincerely,

William

William Teale, PhD
Editor
The EMBO Journal
w.teale@embojournal.org
